# Pregnancy-acquired memory CD4$^+$ regulatory T cells improve pregnancy outcome in mice

Kristin Thiele [1,2] ✉, Christopher Urbschat [1,2], Julia. Isabel. Amambay Riquelme[1], Lisa. Sophie Ahrendt[1], Ronja Wöhrle[1], Steven Schepanski [1,3], Judith Joana Eckert[1,4], Etienne Becht [5,6], Minyue Qi [7], Malik Alawi [7], Martin Becker [8], Nicola Gagliani [2,9,10], Hans-Willi Mittrücker [11], Anke Diemert[12,13] & Petra Clara Arck [1,2,13] ✉

Subsequent pregnancies are generally less prone to obstetric complications. A successful pregnancy outcome requires pivotal immunological adaptation to ensure immune tolerance towards the foetus. Thus, the lower risk for pregnancy complication during subsequent pregnancies may be attributable to immune memory mounted during first pregnancies. Here we identify higher frequencies of fetal-antigen-specific CD4$^+$ regulatory T (Treg) cells both postpartum and in subsequent pregnancies in mice which are partly originating from trans-differentiated Th17 cells. Our functional experiments demonstrate that these CD4$^+$ Treg cells have memory functions (CD4$^+$ mTreg) and account for an improved fetal development and pregnancy outcome, also during adverse conditions, such as gestational sound stress. Using a high-throughput single-cell quantification method, we identify candidate markers for the detection of CD4$^+$ mTreg cells, which include CXCR4 and CD274. Our findings thus contribute to the improved understanding of pregnancy-induced immune memory and foster the identification of immune targets aiming to reduce the risk for immune-mediated pregnancy complications.

Pregnancy success is critically regulated by specific maternal immune and endocrine adaptions. These adaptations allow the semi-allogenic fetus to evade rejection by maternal immune effector cells and develop until term[1]. CD4$^+$ regulatory T (Treg) cells are a critical feature of the maternal immune adaptation by establishing an anti-inflammatory environment locally as well as systemically. CD4$^+$ Treg cells limit excessive inflammation and suppress effector T (Teff) cell reactions in mammalian pregnancies[2–10]. These functions are

[1]Division of Experimental Feto-Maternal Medicine, Department of Obstetrics and Fetal Medicine, University Medical Center Hamburg-Eppendorf, Hamburg, Germany. [2]Hamburg Center for Translational Immunology, University Medical Center Hamburg-Eppendorf, Hamburg, Germany. [3]Institute of Developmental Neurophysiology, Center for Molecular Neurobiology Hamburg (ZMNH), University Medical Center Hamburg-Eppendorf, Hamburg, Germany. [4]Human Development and Health, Southampton General Hospital, University of Southampton, Southampton, UK. [5]Vaccine and Infectious Disease Division, Fred Hutchinson Cancer Research Center, Seattle, USA. [6]Centre de recherche sur l'inflammation, INSERM U1149, Université Paris Cité, Paris, France. [7]Bioinformatics Core, University Medical Center Hamburg-Eppendorf, Hamburg, Germany. [8]Chair for Intelligent Data Analytics, Institute for Visual and Analytic Computing, Department of Computer Science and Electrical Engineering, University of Rostock, Rostock, Germany. [9]Section of Molecular Immunology and Gastroenterology, I Department of Medicine, University Medical Center Hamburg-Eppendorf, Hamburg, Germany. [10]Department of General, Visceral and Thoracic Surgery, University Medical Center Hamburg-Eppendorf, Hamburg, Germany. [11]Institute for Immunology, University Medical Center Hamburg-Eppendorf, Hamburg, Germany. [12]Department of Obstetrics and Fetal Medicine, University Medical Centre Hamburg-Eppendorf, Hamburg, Germany. [13]German Center for Child and Adolescent Health, Hamburg, Germany. ✉e-mail: k.thiele@uke.de; p.arck@uke.de

mediated by secretion of interleukin (IL)−10 and transforming growth factor beta (TGF-β), as well as cytotoxic T-lymphocyte-associated protein 4 (CTLA4) and Programmed death-ligand 1 (PD-L1)-dependent cell-cell interactions[11–17]. Further, CD4[+] Treg cells have been shown to exhibit a distinct phenotypic plasticity to transdifferentiate into T helper (Th)17 cells, e.g., in autoimmune settings[18].

A wealth of observational studies highlights the increased risk of pregnancy complications if a previous pregnancy was already affected by e.g., spontaneous abortion, preeclampsia, preterm birth and others[19–25]. Conversely, biological pathways through which a normally progressing first pregnancy reduces the risk for gestational complications in subsequent pregnancies are still sparse. We could previously show that an uncomplicated first pregnancy in humans is associated with an increased birth weight, a key proxy for appropriate fetal development, by 4.2% in a subsequent pregnancy. We also observed an overall reduced risk for obstetric complications in subsequent human pregnancies[19]. These clinical observations support the existence of a 'gestational memory', which may be initiated during the first pregnancy.

The ability to generate memory cells is a central feature of the adaptive immune system. Memory T cells generally promote an advanced immune response when encountering the same antigen, e.g., upon pathogen re-exposure or vaccinations[26]. However, in the context of reproduction, evidence for the initiation of immune memory during a first pregnancy is still sparse, although strongly suggested by the above described clinical observations. First experimental evidence have been provided for the existence of memory CD4[+] Treg (CD4[+] mTreg) cells in mice, specifically into the antigen-specificity of these cells[27] and the impact of fetal microchimeric cells for the re-expansion of CD4[+] Treg cells[28]. Since these evidences were based on a mouse model with an immune-dominant I-Ab:2W1S55−68 peptide expressed as a surrogate fetal antigen[29], confirmation under physiological settings in mice is needed. Further, the functional role of CD4[+] mTreg cells, such as a re-expansion-related improved outcome in subsequent pregnancies is still elusive. Furthermore, the identification CD4[+] mTreg cells is still ambiguous and often limited to CD45RO in human and CD44 in mice[30–33].

Less obstetric complication and improved neonatal outcome in subsequent pregnancy is often clinically observed, but gaining evidence from human studies to reveal the underlying mechanisms is challenging as peripheral blood is not suitable research material to assess the local immune environment at the fetal-maternal interface. In the present study, we reveal a causal link between the expansion of CD4[+] Treg frequencies in the uterus and uterus-draining lymph node during second pregnancies and an improved fetal outcome in mice. By utilizing a high-throughput single-cell quantification method, we further contribute to a more clear-cut identification of CD4[+] mTreg cells during gestation which has the potential to identify immune targets aiming to reduce the risk for immune-mediated pregnancy complication.

## Results

### Less obstetric complications and an improved fetal growth in subsequent human pregnancies

Within the prospective longitudinal pregnancy cohort study PRINCE (Prenatal Identification of Children's Health), conducted at the University Medical Center Hamburg-Eppendorf, we have 78 women who participated with two consecutive pregnancies (Fig. 1a). Maternal age is naturally significantly higher in second compared to first pregnancy (Fig. 1b), but maternal BMI determined in the first trimester was similar in both pregnancies (Fig. 1c). We further observed a strong trend towards less obstetric complications in second pregnancy (Fig. 1d). Estimated fetal weight assessed via fetal ultrasound in second and third trimester indicated an improved intra-uterine growth in second pregnancies, but levels of significance were not reached (Fig. 1e, mean

±SD of 1st versus (vs) 2nd pregnancy: 1st trimester: 103.0 g ± 21.8 g vs 98.80 g ± 13.43 g, 2nd trimester: 647.5 g ± 77.9 g vs 660.8 g ± 77.1 g, 3rd trimester: 2625 g ± 322.4 g vs 2706 g ± 356.1 g). At birth, we observed a 2.5% increased birth weight in a second pregnancy (Fig. 1f, g) while gestational length was similar (Fig. 1h). Additionally, second pregnancy neonates exhibit a higher Apgar score at 5 minutes suggesting an improved physical and adaptational health, albeit level of significance was not reached (Fig. 1i). PBMCs obtained from 15 women from each trimester revealed frequencies of CD4 Treg cells as well as their CD73 expression to be similar throughout first and second pregnancy (Fig. 1j, k, and Supplementary Fig. 1). Taken together these clinical data support the observation of less complicated subsequent human pregnancies[19]. However, we did not observe any changes in systemic CD4[+] Treg cell frequencies of human PBMCs. Hence, to suitably study the impact and functional relevance of immune memory, we chose various mouse models to investigate the prevailing local immune environment.

### Higher frequency of CD4[+] Treg cells post-partum with increasing number of pregnancies

Age-matched C57Bl6/J female mice were allogenically mated once, twice or three times, respectively. Four weeks after the last delivery, parous mice were compared to virgin mice, that have been left undisturbed during the entire time period (Fig. 2a). Equally to our observation in human PBMCs, we detected similar CD4[+] Treg cell frequencies in murine PBMCs four weeks postpartum, independent on the number of pregnancies. (Fig. 2b, Supplementary Fig. 2). In contrast, with increasing number of pregnancies, we detected higher CD4[+] Treg cell frequencies in the uterus-draining lymph (Fig. 2c, d). Additionally, higher frequencies of CD4[+] Treg cells with a memory phenotype, defined by CD44[high] expression, could by detected with increasing numbers of pregnancies, compared to virgin mice (Fig. 2e, f). Whilst higher CD4[+] Treg cells frequencies could also be detected in the uterus (Fig. 2g, h), the frequencies of CD44[high] CD4[+] Treg cells here remained unaffected by the increasing number of pregnancies (Fig. 2i, j).

However, insights into the generation and functional role of CD4[+] Treg cells with a memory phenotype during pregnancy are still limited from these postpartum analyses.

### Higher levels of CD4[+] Treg cells during second pregnancies

To overcome this limitation, we next analyzed the maternal immune response during first and second pregnancies using Fir/Tiger mice (Fig. 3a). Similar to our post-partum analysis, we could not observe significant differences in PBMCs, spleen and inguinal lymph node during pregnancy, exemplary shown for gestation day (gd) 3.5 in Supplementary Fig. 3a. Consequently, we focused our analysis on key early, mid and late gestational time points of the uterus and the respective draining lymph nodes. We detected higher CD4[+] Treg frequencies in lymph nodes throughout gestation in mice during second pregnancies, compared to first pregnancies (Fig. 3b). A peak of IL-10[+] Treg cells could be identified early in second pregnancies, on gd 3.5 (Fig. 3c, d). Similar to the lymph node, higher CD4[+] Treg cell frequencies were also detected throughout second pregnancies in the uterus (Fig. 3e), along with increased IL-10 expression on gd 3.5 (Fig. 3f, g). Interestingly, maternal serum progesterone levels were unaltered between first and second pregnancies (Fig. 3h), suggesting that the higher CD4[+] Treg cell frequencies in second pregnancies are not attributable to higher levels of progesterone[34,35]. Consequently, the beneficial effects observed during subsequent pregnancy are based on more cell-mediated mechanisms that happen hormone-independent.

During pregnancy, the increase of CD4[+] Treg cells is known to also result from the interaction with IL-10[+] CD80/86[low] tolerogenic dendritic cells (DC)[6,36,37]. Thus, we also characterized uterine DCs and observed a higher IL-10 expression before and during second pregnancies (Fig. 3i, j), whilst the expression of non-tolerogenic co-

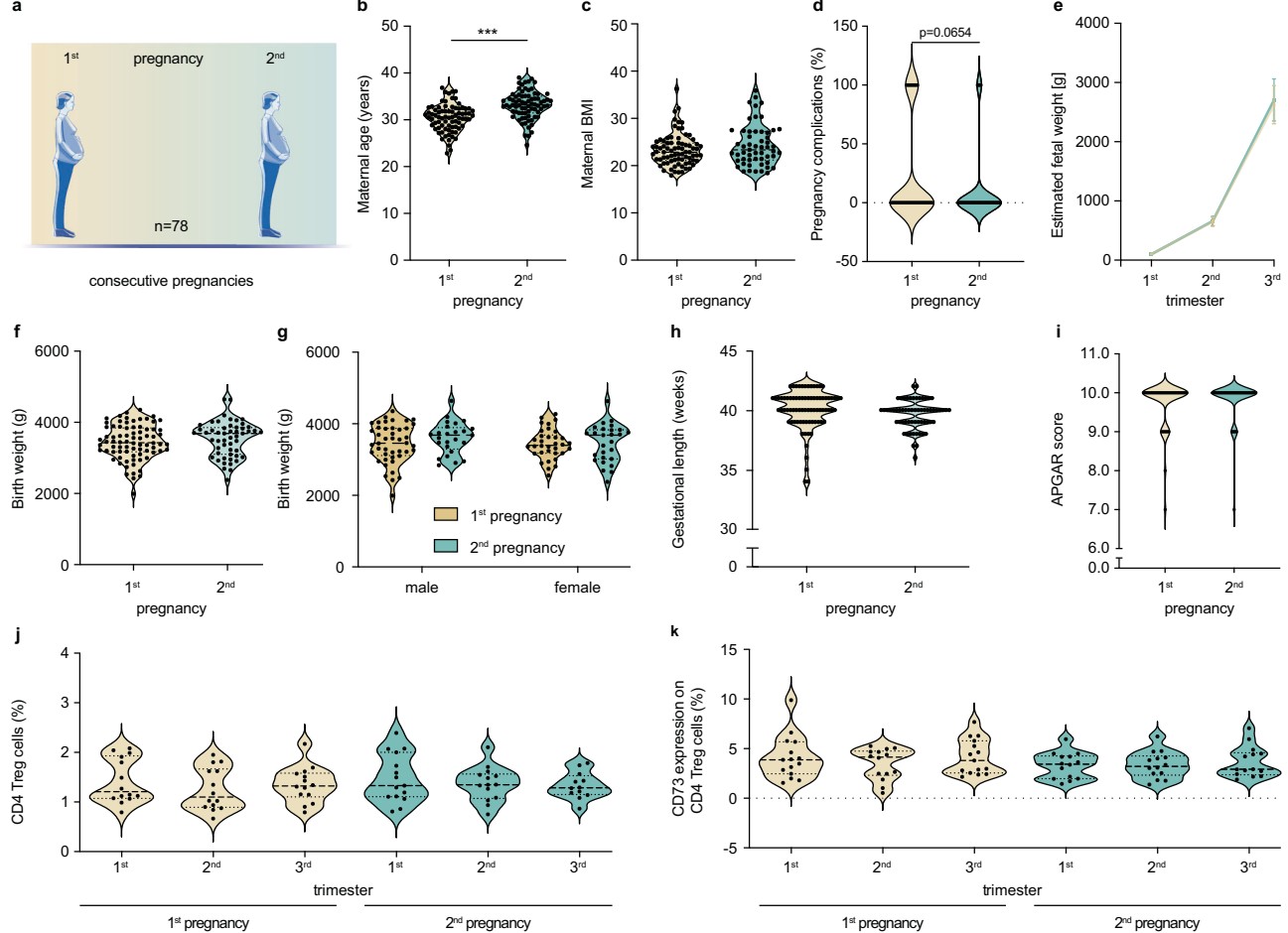

**Fig. 1 | Less obstetric complications and an improved fetal growth in subsequent human pregnancies. a–i** Maternal and fetal data from pregnant women in their first and second pregnancy, respectively, enrolled in the prospective birth cohort study PRINCE ($n = 78$) conducted at the University Medical Center Hamburg-Eppendorf including (**b**) maternal age ($1^{st}$: $n = 78$, $2^{nd}$: $n = 77$, $p = 0.0004$), **c** maternal BMI ($1^{st}$: $n = 78$, $2^{nd}$: $n = 55$), **d** pregnancy complication ($1^{st}$: $n = 54$, $2^{nd}$: $n = 54$), **e** estimated fetal weight assessed via prenatal ultrasound ($1^{st}$ trimester: $1^{st}$: $n = 46$, $2^{nd}$: $n = 46$; $2^{nd}$ trimester: $1^{st}$: $n = 48$, $2^{nd}$: $n = 48$); $3^{rd}$ trimester: $1^{st}$: $n = 47$, $2^{nd}$: $n = 46$), **f** birth weight ($1^{st}$: $n = 78$, $2^{nd}$: $n = 52$), **g** birth weight stratified by neonatal sex (male: $1^{st}$: $n = 44$, $2^{nd}$: $n = 25$; female: $1^{st}$: $n = 34$, $2^{nd}$: $n = 27$), **h** gestation length ($1^{st}$:

$n = 78$, $2^{nd}$: $n = 61$) and **i** APGAR score at 5 minutes ($1^{st}$: $n = 51$, $2^{nd}$: $n = 52$). **j, k** Flow cytometry analysis of PBMCs obtained from pregnant women ($n = 15$) of each trimester in their $1^{st}$ and $2^{nd}$ pregnancy, including frequency of CD4$^+$ Treg cells (**j**) along with their CD73 expression (**k**). **b–d**, **f–k** Data are presented as violin plots with individual point and median and quartiles. **e** Data are presented as mean values ±SEM. The statistical significance was calculated using Student's t-test for comparing two groups (**b–f**, **h**, **i**), Two-way-Anova (**g**) and One-way-ANOVA (**j**, **k**). See Supplementary Fig. 1 for gating strategy. Source data are provided as a Source Data file.

stimulatory molecules such as CD80/86 was significantly lower during subsequent pregnancy (Fig. 3k).

### Improved pregnancy outcome in second pregnancies

On the day of birth, we observed an increased litter size born to females upon second and third pregnancies, compared to first pregnancy litters (Fig. 4a), suggesting fewer abortion in utero. A second indicator of an improved pregnancy outcome was an increased neonatal weight, particularly in third pregnancies (Fig. 4b). These observations in mice mirror the observations in humans of an improved neonatal outcome in subsequent pregnancies[19].

We also assessed key parameter of fetal outcome, e.g., the expression of the transcription factors Sox2 and Nanog, which are indicative for timely fetal development early during pregnancy, on gd 7.5, by confocal microscopy[38,39]. Here, we observed higher expression levels of Sox2 and Nanog in foetuses in second, compared to first pregnancies, albeit only significant for SOX-2 (Fig. 4c–e) which may indicate an advanced early fetal development. At time points later in gestation, the abortion rate was significantly lower in second pregnancies (Fig. 4f), along with a higher fetal weight (Fig. 4g).

Interestingly, we also detected larger placentas later during gestation in second pregnancies (Fig. 4h), which resulted from the enlargement of both, the fetal labyrinth as well as the junctional zone (Fig. 4i–k), albeit the changes were only significant for the latter.

### Modulation of CD4$^+$ Treg cell number and antigen-specificity during pregnancies

In order to understand the causal role of the CD4$^+$ Treg cells we detected at higher frequencies in second pregnancies, we utilized the DEpletion of REGulatory T cells (DEREG) mouse model (Fig. 5a). We established a dose of diphtheria toxin (DT) for sufficient CD4$^+$ Treg cell depletion during the interpregnancy interval (Supplementary Fig. 4a), Further, we monitored the recovery of the CD4$^+$ Treg cells to ensure normal CD4$^+$ Treg cell frequencies before the second mating (Supplementary Fig. 4b). Additionally, to exclude side effects of DT on our analyzed parameters, we performed a control experiment using C57B/6 J mice and did not observed any alterations (Supplementary Fig. 4c). The depletion of CD4$^+$ Treg cells after successful completion of the first pregnancy resulted – as expected – in lower CD4$^+$ Treg frequencies in uterus-draining lymph

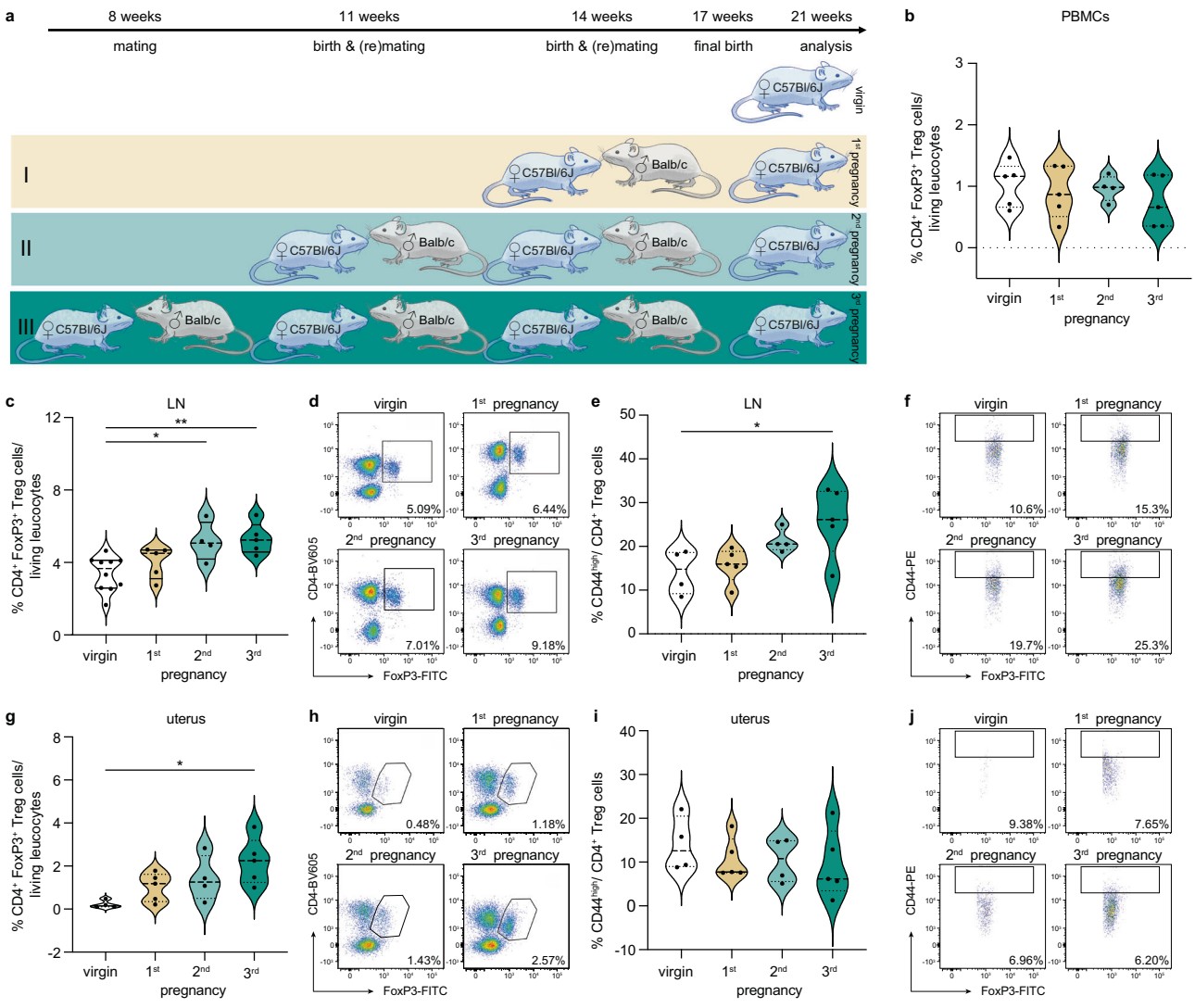

**Fig. 2 | Higher frequency of CD4⁺ regulatory T (Treg) cells post-partum with increasing number of pregnancies. a** Experimental setup: age-matched C57Bl/6 J mice were allogenically mated to Balb/c males once, twice or three times, respectively, and compared to virgin mice at the age of 21 weeks. **b–j** Flow cytometry analysis was performed 4 weeks after the last delivery including frequencies of CD4⁺ Treg cells and associated CD44$^{high}$ expression in (**b**) PBMCs (virgin: $n = 5$; 1st: $n = 5$, 2nd: $n = 4$, 3rd: $n = 5$), **c–f** lymph node (virgin: $n = 10$ (4); 1st: $n = 5$, 2nd: $n = 4$, 3rd: $n = 5$; **c** virgin vs 2nd: $p = 0.0277$, virgin vs 3rd: $p = 0.0078$; **e** virgin vs 3rd: $p = 0.0388$) and (**g–j**) uterus (virgin: $n = 4$; 1st: $n = 5$, 2nd: $n = 4$, 3rd: $n = 5$; **g** virgin vs 3rd: $p = 0.0194$), respectively. Additional plots are shown in Supplementary Fig. 2. Data are presented as violin plots with individual point and median and quartiles. The statistical significance was calculated using One-way-Anova (* $p < 0.05$, ** $p < 0.01$). Source data are provided as a Source Data file.

node and uterus during second pregnancies (Fig. 5b–d, Supplementary Fig. S5a, b).

Subsequently, we adoptively transferred first time pregnant mice (sired by ovalbumin (OVA)-males) with CD4⁺ Treg cells isolated from either virgin or parous female mice. Here, the parous donor mice had also been mated to OVA-males, resulting in the placental expression of OVA and thus, fetal antigen specificity of CD4⁺ Treg cells (Fig. 5e). In a separate set of experiments, we could indeed confirm higher numbers of fetal antigen (OVA)-specific CD4⁺ Treg cells in second pregnancies (Fig. 5f). Interestingly, adoptive transfer of CD4⁺ Treg cells from parous donors led to a higher frequency of uterine CD4⁺ Treg cells in first time pregnant mice, whilst frequencies in the uterus-draining lymph node remained similar between groups (Fig. 5g, h).

In order to further assess the fetal antigen specificity of CD4⁺ Treg cells mounted during first pregnancies in mice, we designed experiments in which female mice were mated to a different male strain to induce the second pregnancy (Fig. 5i). Here, we detected lower CD4⁺ Treg cell frequencies in second pregnancies if sired by a different male

(Fig. 5j–l), whereby no differences in CD44$^{high}$ or IL-10 expression could be identified (Supplementary Fig. 5c–f).

The frequency of CD4⁺ Treg frequencies can be affected by adverse conditions, e.g., gestational sound stress, which also impairs fetal outcome[40,41]. When exposing first- and second-time pregnant mice to sound stress (Fig. 5m), we observed that CD4⁺ Treg cell frequencies were refractory to such challenge during second pregnancies, at least in the uterus draining lymph nodes (Fig. 5n–p). A side observation we made was that progesterone levels were significantly higher upon sound stress in subsequent pregnancies (Supplementary Fig. 5g).

## Consequences of CD4⁺ Treg cell modulation on pregnancy outcome

Next, we assessed the pregnancy outcome triggered by the experimental modulation of CD4⁺ Treg cell number and antigen-specificity described above (Fig. 5). The depletion of CD4⁺ Treg cells during the interpregnancy interval (see also Fig. 5a) resulted in a significant

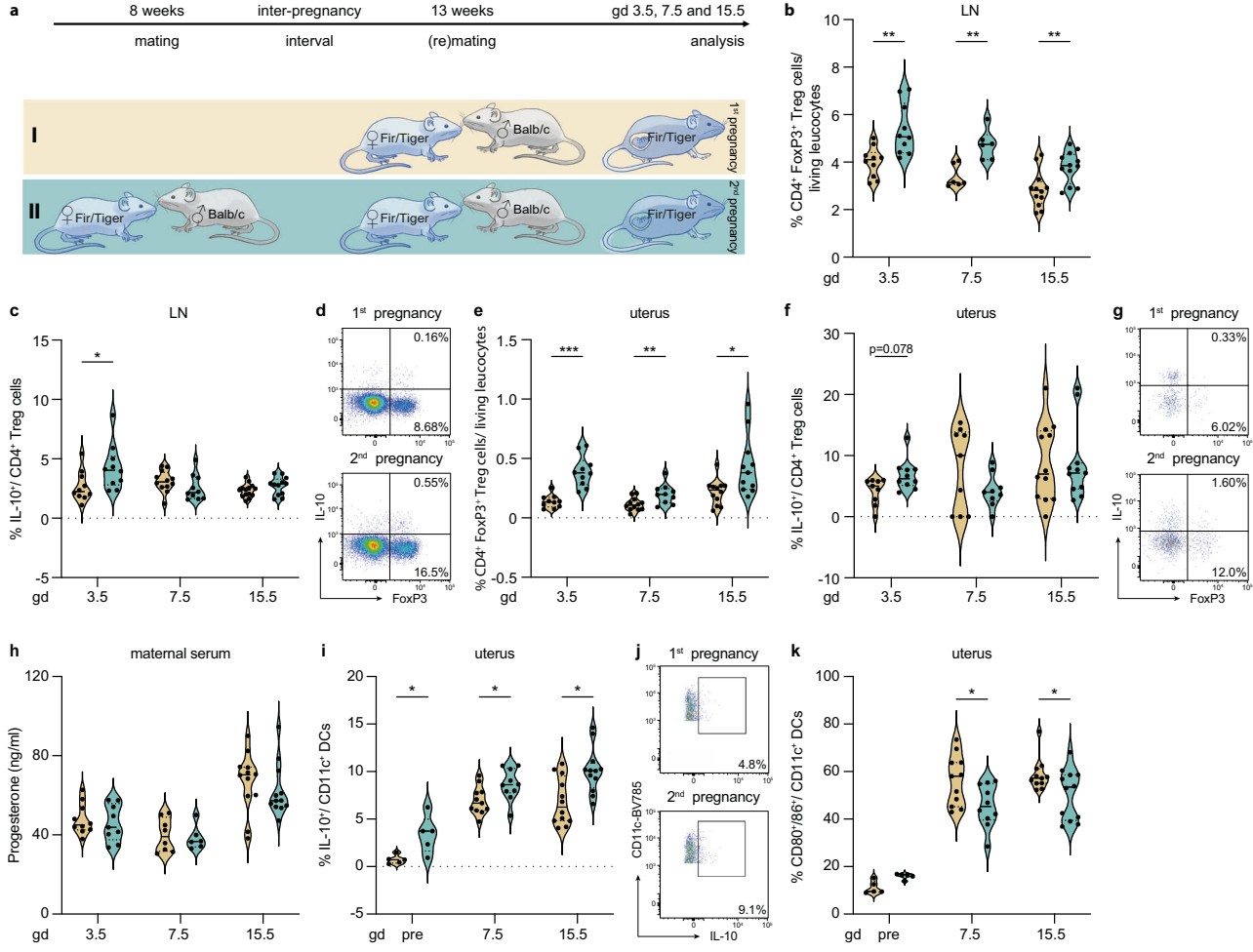

**Fig. 3 | Higher levels of CD4+ regulatory T (Treg) cells during second pregnancies. a** Experimental setup: age-matched Fir/Tiger mice were allogenically mated to Balb/c males once or twice, respectively, and evaluated on various time points during pregnancy (gd 3.5: n = 10, gd 7.5: n = 6 (10), gd 15.5: n = 12). **b–g** Flow cytometry analysis included frequencies of CD4+ Treg cells and associated ex-vivo IL-10 expression in (**b–d**) lymph node (**b** gd 3.5: p = 0.0028, gd 7.5: p = 0.002, gd 15.5: p = 0.0042; **c** gd 3.5: p = 0.042) and (**e–g**) uterus (**e** gd 3.5: p = 0.000024, gd 7.5: p = 0.0086, gd 15.5: p = 0.011; **f** gd 3.5: p = 0.078), respectively. Pseudocolour plots and respective numbers presented correspond to CD4 as parent population.

Additional plots are shown in Supplementary Fig. 3. **h** Serum progesterone levels were assessed throughout pregnancy. **i–k** Flow cytometry analysis assessed (**i, j**) ex-vivo IL-10 expression (gd 3.5: p = 0.029, gd 7.5: p = 0.048, gd 15.5: p = 0.010) and (**k**) CD80+/CD86+ expression (gd 3.5: p = 0.02, gd 7.5: p = 0.03, gd 15.5: p = 0.04) on CD11c+ dendritic cells (DC). Data are presented as violin plots with individual point and median and quartiles. The statistical significance was calculated using Multiple unpaired t-tests (* p < 0.05, ** p < 0.01, *** p < 0.001). Source data are provided as a Source Data file.

decline of Sox2 and Nanog early during gestation (Fig. 6a), as well as reduced protection from fetal abortions (Fig. 6b left) and lower fetal weight (Fig. 6b right) later during gestation. This was accompanied by smaller labyrinths leading to a reduced placental ratio in second pregnancies treated with DT (Supplementary Fig. 6a–e). Adoptive transfer of CD4+ Treg cells from virgin and parous donors (see also Fig. 5e) only marginally affected pregnancy outcome parameter such as fetal abortions or weight (Fig. 6c). Interestingly, the importance of the antigen-specificity of CD4+ Treg cells (see also Fig. 5i) was highlighted by our observation of an increased frequency of fetal abortions when a second pregnancy was sired by a different parental strain (Fig. 6d left). Here, the fetal weight was not affected by the paternal strain (Fig. 6d right). The higher frequencies of CD4+ Treg cells we detected in second pregnancies in response to sound stress (see also Fig. 5m) did not interfere with the abortion rates (Fig. 6e left), as expected due to the late stress exposure during gestation we chose. However, fetuses during second pregnancies were less affected by a stress-induced fetal growth restriction, mirrored by a significantly lower fetal weight in first pregnancy which was not present in second pregnancy (Fig. 6e right). Further, we observed improved placental

features in second pregnancy, that were not affected by prenatal stress challenge (Supplementary Fig. 6f–h).

## Plasticity of CD4+ Treg cells in first and second pregnancies

CD4+ Treg cells can transdifferentiate from Th17 cells, e.g., during the resolution of inflammation[42]. Since implantation and induction of labor are inflammatory events, we next tested if CD4+ Treg cells derive from Th17 cells during pregnancy (exIL-17). Therefore, we used the Fate mouse model, a combination of a IL-17A fate reporter mouse[43] with a IL-17A, IL-10, Foxp3 triple reporter mouse model[42,44,45]. Cells that have previously expressed high level of *Il17a* are permanently marked by YFP and can be investigated for present IL-17A, IL-10 and Foxp3 expression ex vivo without in vitro restimulation (Fig. 7a). As expected, exIL-17 cells in virgin mice were virtually non-existent, but detectable at significantly higher frequencies at different timepoints in second pregnancies compared to first pregnancies (Fig. 7b). These exIL-17 cells either transdifferentiated into CD4+ Treg cells – partly expressing IL-10 – or into type 1 regulatory T (Tr1) cells, which lack the FoxP3 (Fig. 7c). In this set of experiments, we could independently confirm the previously observed higher CD4+ Treg cell frequencies in second

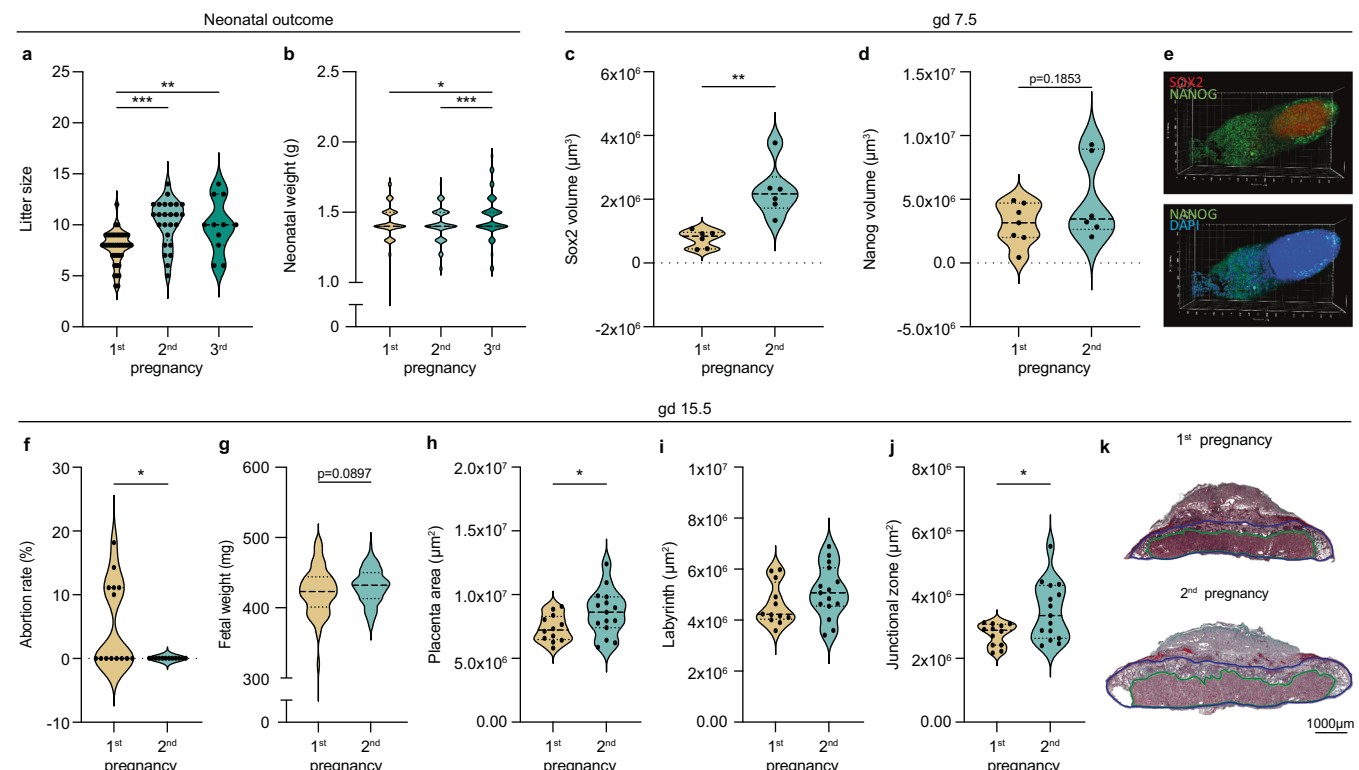

**Fig. 4 | Improved pregnancy outcome in second pregnancies.** Reproductive outcome including **a** litter size (1st: $n = 41$, 2nd: $n = 25$, 3rd: $n = 11$; 1st vs 2nd: $p = <0.0001$, 1st vs 3rd: $p = 0.0081$) and (**b**) neonatal weight (1st: $n = 320$, 2nd: $n = 253$, 3rd: $n = 112$; 1st vs 3rd: $p = 0.0262$, 2nd vs 3rd: $p = 0.0004$) was assessed on day of birth. Further, fetal growth and development were evaluated on gestation day (gd) 7.5 and 15.5. Fetuses from gd 7.5 (1st: $n = 7$, 2nd: $n = 6$) were stained against (**c**) Sox3 (1st vs 2nd: $p = 0.0016$) and (**d**) Nanog to assess early fetal development. Representative pictures are shown in (**e**). On gd 15.5, pregnancy outcome was assessed including (**f**) abortion rate (1st: $n = 14$, 2nd: $n = 12$; 1st vs 2nd: $p = 0.0108$, **g** fetal weight (1st: $n = 115$, 2nd: $n = 106$) and **h-j**

placental histomorphology (1st: $n = 12$, 2nd: $n = 15$; **h** 1st vs 2nd: $p = 0.0463$; **j** 1st vs 2nd: $p = 0.0133$. **k** Representative photomicrographs illustrating mid-sagittal sections of gd 15.5 placental tissue from first (top) and second (bottom) pregnancy. Black line in the picture denotes 1000 μm, green lines encircle the labyrinth, blue lines surround the junctional zone. Data are presented as violin plots with individual point, median and quartiles, and the statistical significance between first and second pregnancy was calculated using One-way-Anova (comparing three group) or Student's t-test (comparing two groups), respectively (* $p < 0.05$, ** $p < 0.01$, *** $p < 0.001$). Source data are provided as a Source Data file.

pregnancy also in Fate females (Fig. 7d). Additionally, we could demonstrate that ~2% of these CD4+ Treg cells in second pregnancy were either exIL-17 cells or CD4+ Treg cells that had previously produced IL-17 (Fig. 7e). Also, Tr1 cells exhibited significantly increased frequencies in second pregnancies, compared to first pregnancies, whilst overall frequencies were very low (Fig. 7f). However, in contrast to CD4+ Treg cells, the frequency of exIL-17 cells within the Tr1 cell compartment remained unchanged between first and second pregnancy (Fig. 7g). Higher CD4+ Treg cell levels in second pregnancy of Fate females were also confirmed for the uterus (Supplementary Fig. 7a) whilst alterations of fate markers were only marginal, likely due to the low cell numbers (Supplementary Fig. 7b–d). Additionally, the effects of a prenatal stress challenge on CD4+ Treg cell frequencies (Fig. 5n–p) and pregnancy outcome (Fig. 6e) could be independently confirmed using the fate mice (Supplementary Fig. 8a–d).

### Immune phenotyping of CD4+ Treg cells from virgin and parous mice

Given the observed pregnancy outcome and the functional importance of CD4+ Treg cells for reproductive success, we next aimed to identify reliable and more specific immune markers to improve the detection of CD4+ Treg cells generated during pregnancy. We used high-throughput flow cytometry and subsequently analyzed the data by Infinity Flow[46] and a direct cell set approach. (Fig. 8a).

Within the CD3+ T cell compartment derived from virgin and parous mice, we identified four cell clusters which include CD8+ T cells, γδ T cells, CD4+ Treg and non-Treg (CD4+ conventional T (Tcon)) cells

(Fig. 8b). As expected from our conventional flow cytometry analyses (see Fig. 2c), the proportion of CD4+ Treg cells in parous mice was visibly larger compared to virgin mice. Subsequent to machine learning based prediction of the single cell expression of the Infinity panel markers using Infinity Flow[46], a discrimination score was calculated indicating the ability of each marker to distinguish between the two conditions based on an expression vector (Fig. 8c). Hereby, we identified twelve markers that were differently expressed on CD4 Treg cells of parous mice ($0.25 > AUC > 0.75$). Among these markers were CD45.2 and Ig light chain κ, which we excluded from further analysis as CD45.2 is an alloantigen of CD45 and expressed by Ly5.2 bearing mouse strains (e.g., C57Bl/6 J), which were used in this study, and Ig light chain κ is a small polypeptide subunit of antibodies and presumably cross-reacted with the mouse antibodies used in the backbone panel. Subsequently, we retraced the flow cytometry data and calculated frequency and mean fluorescence intensity (MFI) for the predicted markers presumed to distinguish between CD4+ Treg cells originating from virgin and parous mice, excluding three markers with a positive frequency below 5% in both groups. CD184 (C-X-C chemokine receptor type (CXCR)-4), CD274 (PD-L1) and CD81 were identified as the markers with the highest frequency (Fig. 8d).

To supplement this analysis based on Infinity Flow, the cell subsets for each marker were analyzed directly. Here, marker selection was primarily based on the effect size Cohen's d (Fig. 8e), which allows identifying maximally discriminating marker distributions. Hereby, we identified 15 markers that were up-regulated in CD4+ Treg cells derived from parous mice and four markers that were down-regulated as

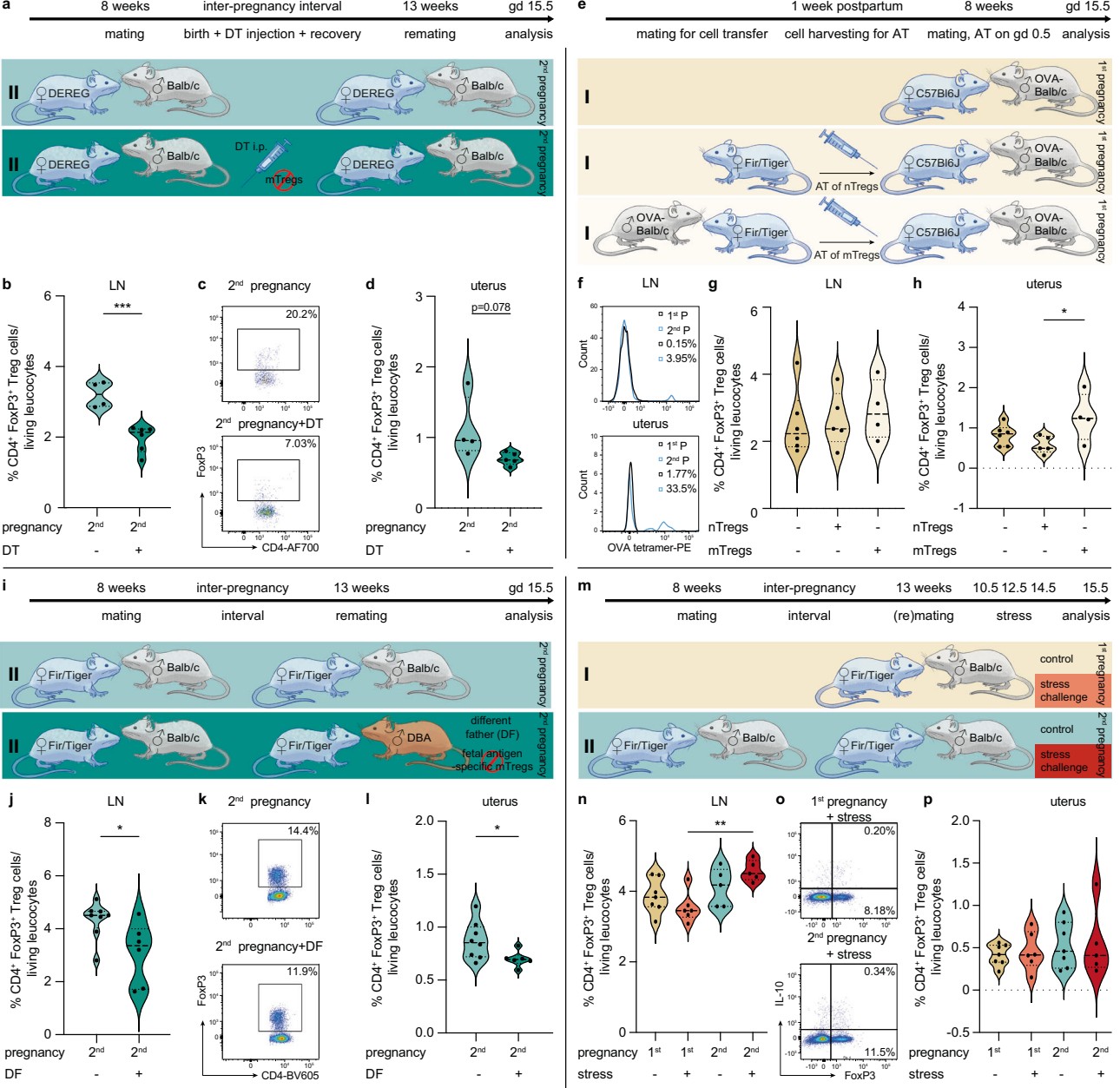

**Fig. 5 | Modulation of CD4⁺ regulatory T (Treg) cell number and antigen-specificity during pregnancies. a** Age-matched DEREG mice (2nd: $n = 4$, 2nd + DT: $n = 6$) were allogenically mated to Balb/c males twice. Some second pregnancy mice were injected with diphtheria toxin (DT) after the first delivery to deplete CD4⁺ Treg cells. On gestational day (gd) 15.5, frequencies of CD4⁺ Treg cells in (**b, c**) lymph node (2nd vs 2nd + DT: $p = 0.0008$) and (**d**) uterus were assessed by flow cytometry. **e** 150,000 CD4⁺ Treg cells – either harvested from virgin mice (termed naïve (n) Tregs) or from parous mice (termed memory (m)Tregs) – were adoptively transferred (AT) into first pregnancy C57Bl6/J mice mated to OVA-Balb/c males on the day of plug (1st: $n = 6$, 1st+nTregs: $n = 5$, 1st+mTregs: $n = 4$). **f** Histograms of fetal antigen (OVA)-specific CD4⁺ Treg cells in first (black line) and second (blue line) in lymph node (LN, top) and uterus (bottom). Numbers represent the frequency of OVA⁺ cells in the CD4⁺ Treg cell compartment. Flow cytometry analysis of CD4⁺ Treg frequencies was performed in (**g**) lymph node and (**h**) uterus (1st+nTregs vs

1st+mTregs: $p = 0.0403$). **i** Age-matched Fir/Tiger mice were allogenically mated to Balb/c or DBA male mice for their second pregnancy (DF = different father; 2nd: $n = 8$, 2nd + DF: $n = 6$), respectively. On gd 15.5, frequencies of CD4⁺ Treg cells in (**j, k**) lymph node (2nd vs 2nd + DF: $p = 0.0279$) and (**l**) uterus (2nd vs 2nd + DF: $p = 0.0456$) were assessed by flow cytometry. **m** Age-matched Fir/Tiger mice were mated to Balb/c males once or twice and prenatally sound-stressed mid-gestationally (1st: $n = 7$, 1st+stress: $n = 6$, 2nd: $n = 5$, 2nd+stress: $n = 5$). On gd 15.5, flow cytometry analysis assessed CD4⁺ Treg cell frequencies in (**n, o**) lymph node (1st+stress vs 2nd+stress: $p = 0.0071$) and (**p**) uterus. Data are presented as violin plots with individual point, median and quartiles, and the statistical significance between groups was calculated using Student's t-test when comparing two groups or One-way ANOVA when comparing three or more groups (* $p < 0.05$, ** $p < 0.01$, *** $p < 0.001$). Additional data are provided in Supplementary Fig. 4 and 5. Source data are provided as a Source Data file.

indicated by the fold change (Fig. 8f). The highest fold change was observed for CD38 followed by CD20 and CXCR4. Additionally, folate receptor (FR) 4, CD73 and Ly-6A/E display a high frequency and an increased MFI in CD4⁺ Treg cells from parous mice (Fig. 8g).

In summary, both bioinformatic analyses share five markers as potential candidates to distinguish CD4⁺ Treg cells from virgin and parous mice, respectively. These include programmed PD-L1, CXCR4, CD20, CD80 and CD81 (Fig. 8h).

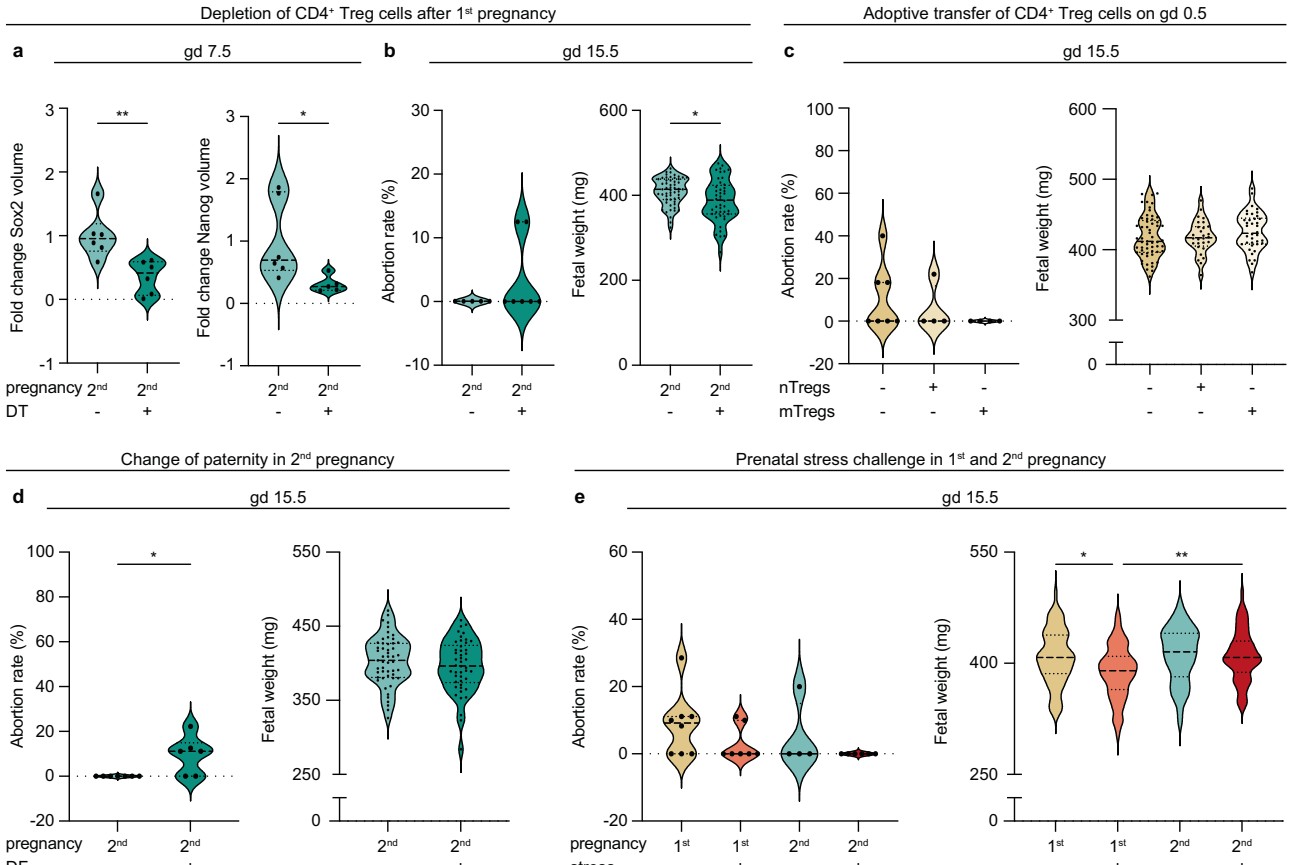

**Fig. 6 | Consequences of CD4⁺ regulatory T (Treg) cell modulation on pregnancy outcome.** Fetal outcome following Diphtheria toxin (DT)-induced depletion of CD4⁺ Treg cells was determined by (**a**) expression of Sox2 ($2^{nd}$: $n = 6$, $2^{nd} + DT$: $n = 6$, $p = 0.0052$) and Nanog ($2^{nd}$: $n = 6$, $2^{nd} + DT$: $n = 5$, $p = 0.0434$) on gestation day (gd) 7.5 as well as (**b**) abortion rate ($2^{nd}$: $n = 5$, $2^{nd} + DT$: $n = 7$) and fetal weight ($2^{nd}$: $n = 55$, $2^{nd} + DT$: $n = 44$, $p = 0.0115$) on gd 15.5, placental histomorphology data are shown in Supplementary Fig. 6. Abortion rate and fetal weight were further assessed on gd 15.5 (**c**) in response to adoptively transferred naïve CD4⁺ (n)Treg cells and memory CD4⁺ (m)Treg cells harvested from virgin and parous mice, respectively, into first pregnancy mice mated to OVA-Balb/c males on the day of plug ($1^{st}$: $n = 7$ (59 fetuses), $1^{st} + nTregs$: $n = 4$ (27 fetuses), $1^{st} + mTregs$: $n = 4$ (37 fetuses)), **d** in response to change in paternity by mating with a different mouse strain (DBA/J instead of Balb/c) for the second pregnancy (DF = different father, ($2^{nd}$: $n = 7$ (52 fetuses), $2^{nd} + DF$: $n = 6$ (48 fetuses); $2^{nd}$ vs $2^{nd} + DF$: $p = 0.0122$) and (**e**) in response to a prenatal stress challenge on gd 10.5, 12.5 and 14.5 ($1^{st}$: $n = 8$ (68 fetuses), $1^{st}+stress$: $n = 7$ (61 fetuses), $2^{nd}$: $n = 4$ (41 fetuses), $2^{nd}+stress$: $n = 6$ (55 fetuses); $1^{st}$ vs $1^{st}+stress$: $p = 0.0152$, $1^{st}+stress$ vs $2^{nd}+stress$: $p = 0.0085$). Placental histomorphology data after prenatal stress challenge are shown in Supplementary Fig. 6. Please revert to Fig. 5 for more detailed descriptions of the experimental setups. Data are presented as violin plots with individual point, median and quartiles, and the statistical significance between groups was calculated using student's t-test when comparing two groups or One-way-ANOVA when comparing three or more groups (* $p < 0.05$). Source data are provided as a Source Data file.

## Discussion

In the present study, we provide strong evidence that CD4⁺ Treg cell frequencies are increased during the postpartum period and second pregnancies in mice. Our results indicate that these increased CD4⁺ Treg cell frequencies emerge from fetal antigen-specific CD4⁺ mTreg cells. Moreover, a trans-differentiation of CD4⁺ Treg cells from Th17 cells, e.g., upon implantation- or labor-related inflammation during first pregnancies can also be postulated. These higher CD4⁺ Treg cell frequencies are solidly linked to an improved pregnancy outcome, as reflected by a reduction of the abortion rate, an improved fetal development and superior placental features. These findings underscore that CD4⁺ mTreg cells emerge during a first pregnancy and – if unchallenged – protect subsequent pregnancies from challenges and complications. The challenges we tested also included a changed paternity, which abrogated the prevention of abortions along with lower CD4⁺ Treg cell frequencies in a second pregnancy in mice. A similar observation was already shown for the risk of preeclampsia in humans. Hereby, nulliparous women and multiparous women with changed paternity showed an equal risk of 3.2% and 3.0% to develop preeclampsia, respectively, while women with no change of partner display a decreased risk of 1.9%[47]. Hence, it can be concluded that the re-exposure to the same paternal antigens is essential for the formation of an immune response favorable for fetal development and growth.

The generation of mCD4⁺ Treg cells has already been shown in other immunological settings, e.g., in the context of viral infections[11,48,49]. Memory T cells are preferentially located in secondary lymphoid organs, mainly in lymph nodes[50]. Hence, it was not surprising to find an increased presence of CD4⁺ CD44^high mTreg cells in the uterus-draining lymph node with increasing number of pregnancies. However, a similar increase of CD4⁺ mTreg cells could not be detected in the uterus, although specifically tissue-resident memory T ($T_{RM}$) cells have been reported for various mucosal tissue including the skin, the small intestine and also the female genital tract in mice and human[51–53]. $T_{RM}$ cells are generally characterized by the expression of the C-type lectin CD69 and/or the integrin CD103, but especially CD103 was only low expressed on uterine CD4⁺ Treg cells in postpartum mice (Supplementary Fig. 9) suggesting that uterine $T_{RM}$ cells with a regulatory phenotype might exhibit not yet identified characteristics. Further, the ability of CD4⁺ Treg cells to remain tissue

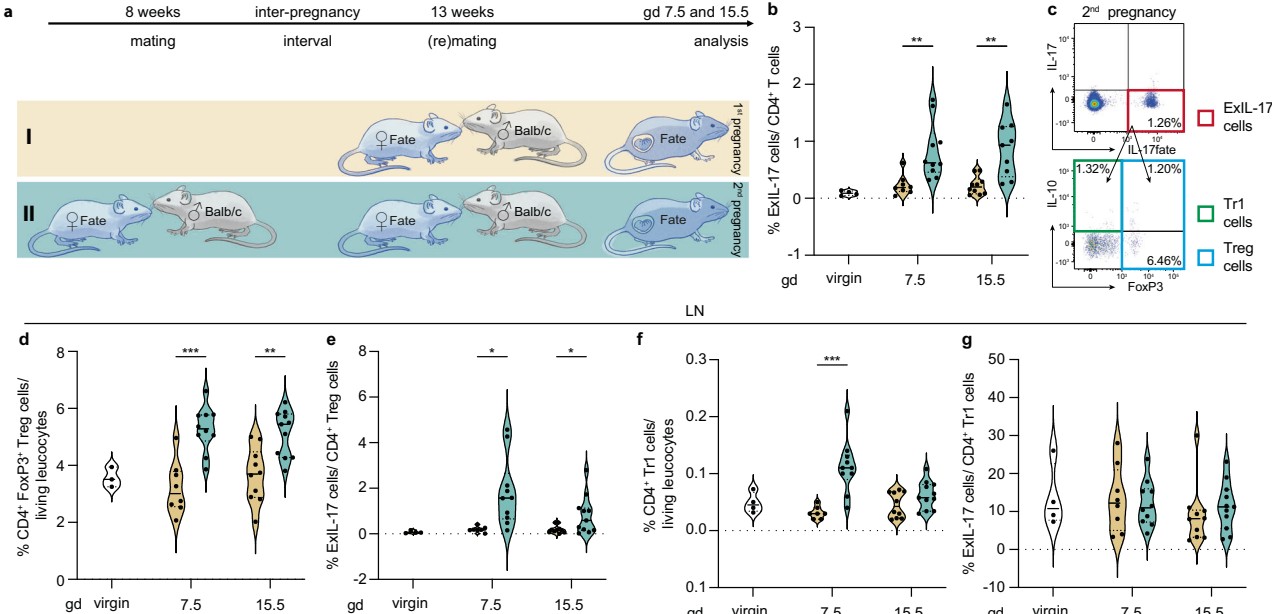

Fig. 7 | Plasticity of CD4⁺ regulatory T (Treg) cells in first and second pregnancies. a Experimental setup: age-matched Fate mice were allogenically mated to Balb/c males once or twice, respectively. b–g Flow cytometric analysis of uterus-draining lymph node harvested on different gestational days (gd) was performed to assess frequency of (b) exIL-17 cells in CD4 T cells (virgin: $n = 4$, gd 7.5: $n = 8$ (1st) and 10 (2nd), gd 15.5: $n = 10$ (1st) and 11 (2nd)) with representative dot plots illustrating c, top: current and former IL-17 producing cells and c, bottom: the acquired phenotype of exIL-17 cells ex-vivo. d Further, CD4⁺ Treg cell frequencies were determined (gd 7.5: p = 0.000216, gd 15.5: $p = 0.00113$) along with (e) the frequency of exIL-17 cells within the CD4 Treg population (gd 7.5: $p = 0.026$, gd 15.5: $p = 0.035$) and (f) Type 1 regulatory T (Tr1) cell frequency (gd 7.5: $p = 0.000413$) in association with (g) the frequency of exIL-17 cells within the Tr1 cell compartment. Data for uterine tissue samples are presented in Supplementary Fig. 7. Additional data regarding independent confirmation of prenatal stress-induced changes in Fate mice are provided in Supplementary Fig. 8. Data are presented as violin plots with individual points, median and quartiles, and the statistical significance between first and second pregnancy was calculated using Multiple unpaired t-tests (* $p < 0.05$, ** $p < 0.01$, *** $p < 0.001$). Source data are provided as a Source Data file.

resident long-term is still rather controversial. So called "tissue Tregs" are predominantly investigated in visceral adipose tissue where they control local inflammation[54]. It was shown that inflammation-experienced "memory" Tregs preferentially localized in non-lymphoid tissue, partially caused by the expression of CXCR3[55]. Further, CD4⁺ Treg cells in the female reproductive tract were found to be more activated compared to circulating CD4⁺ Treg cells, indicated by higher expression of ICOS, TIGIT, CD39, CTLA-4, and GITR, suggesting also higher suppressive capacity[56]. Hence, the interplay of uterine $T_{RM}$ cells and memory CD4⁺ Treg cells might be crucial to balance immune surveillance and fetal tolerance and the functional role of tissue-resident CD4⁺ mTreg cells in ameliorating the outcome of second pregnancy needs further clarification. This includes the actual location of $T_{RM}$ cells in the endometrium or the decidua which could by identified by spatial transcriptomics, mapping the whole transcriptome or targeted gene expression to specific locations in a tissue.

CD4⁺ Treg function are mediated by secretion of cytokines such as IL-10 and TGF-β. Specifically, IL-10 could play a special role here, since elevated IL-10 production in subsequent pregnancies was not only observed in CD4⁺ Treg cells, but also in DCs. Further, that increase was most pronounced in early pregnancy (gd 3.5) which might contribute to the almost complete absence of abortions in subsequent pregnancies as decreased levels of both, CD4⁺ Treg cells and IL-10, are linked to recurrent pregnancy loss[57]. Hence, in further studies it will be important to dissect the impact of CD4⁺ Treg cells from the functional relevance of IL-10 as a potential key molecule by using e.g., IL-10 knockout mice or more specifically creating a conditional knockout of IL-10 on either CD4⁺ Treg cells or DCs.

This further highlights the necessity to identify memory markers of CD4⁺ Treg cells to enable the studying of beneficial effects of CD4⁺ Treg cells in subsequent pregnancy, but also in other immune regulatory settings. We here aimed to dissolve this lack of specific and

reliable memory marker to identify CD4⁺ mTreg cells by employing two bioinformatic methods. We initially limited this analysis to the uterus draining lymph node, as CD4⁺ mTreg cells are preferably located there, but also feasibility of this method is aggravated, because number of immune cells, specifically CD3⁺ T cells in a virgin uterus is low and would require a large number of mice. This interferes with the 3 R principle in animal research (replace, reduce, refine)[58] and thus, will have to be solved in the future, when in-depth analysis of small cell numbers will likely have further advanced.

Among the markers we here identified to distinguish between naïve and memory CD4⁺ Treg cells, CXCR4 and PD-L1 may be most suitable to be included in future studies. The ligand of CXCR4, CXCL12, is abundantly expressed on trophoblast cells. Hence, upon receptor-ligand interaction, CXCR4 may not only regulate CD4⁺ Treg accumulation at the maternal-fetal interface[59], but the consistent up-regulation of CXCR4 on CD4⁺ Treg cells during the interpregnancy interval might facilitate a more rapid re-establishment of fetal tolerance and implantation in early second pregnancy and account for the reduced abortion rate observed in this study. Interestingly, CXCR4^Treg-KO mice display an increased frequency of fetal abortion[60]. Similarly, CXCR4 expression has been shown to be significantly reduced in women with spontaneous miscarriages[61]. The second candidate marker of CD4⁺ mTreg cells we propose, PD-L1, is a ligand of PD-1 and well known to be expressed on CD4⁺ Treg cells. PD-1/PD-1L interaction plays a critical role in regulating T cell tolerance[62], including at the feto-maternal interphase[5,63]. Future experiments should aim at improving our understanding of the role of PD-1/PD-1L in converting Teff cells into Treg cells[64] in the context of pregnancy, since our data indicate a significant contribution of T cell plasticity to enhanced CD4⁺ Treg frequencies in second pregnancy. Further, Tcon cells might be more vulnerable to PD-1/PD-L1 mediated inhibition of effector T-cell differentiation and function and Treg cells might exhibit an enhanced

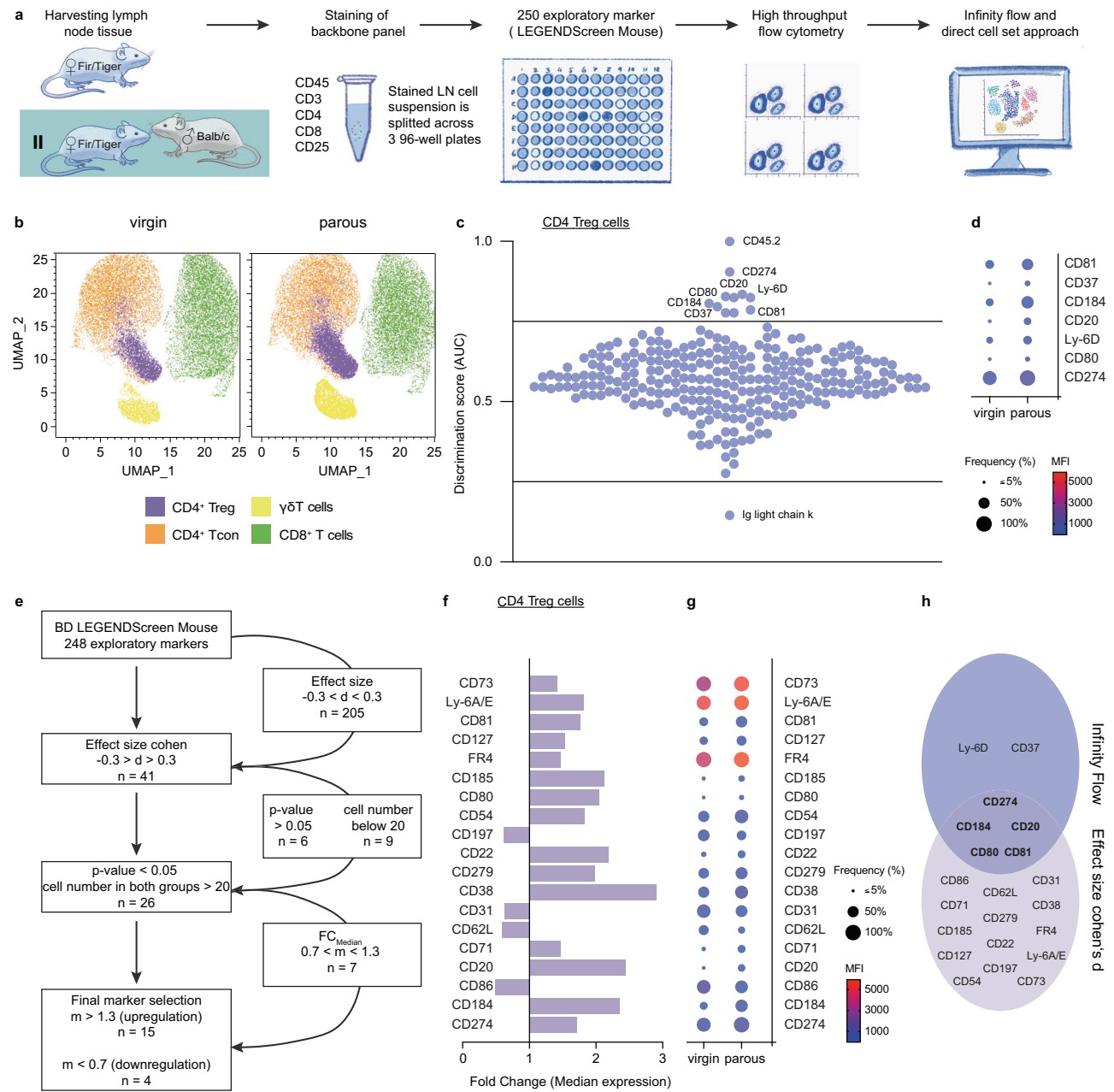

**Fig. 8 | Immune phenotyping of CD4+ regulatory T (Treg) cells from virgin and parous mice. a** Workflow: age-matched Fir/Tiger mice were allogenically mated twice to Balb/c males. Within 7 days postpartum, parous mice (*n* = 3) along with virgin control (*n* = 6) mice were euthanised at the age of 12 weeks, the lymph nodes were harvested, and cells were pooled and stained with the respective backbone panel. Cell suspension was split across 250 marker proteins and subsequently high throughput flow cytometry was performed and the Infinity Flow pipeline and clustering algorithms implemented. **b** UMAP illustrating the respective cell clusters for virgin (left) and parous mice (right). **c** Heatmap displaying the discrimination score for predicted average expression of the Infinity panel marker. Values < 0.25 and > 0.75 were considered suitable to distinguish between the two conditions. **d** frequency and mean fluorescence intensity (MFI) for the seven predicted immune markers. **e** Marker selection by direct cell set approach using effect size Cohen's d. **f** Fold change of median expression was calculated for 19 identified immune marker and frequency and MFI are displayed in **g**. Both analyses shared five markers as potential candidates to distinguish naïve and memory CD4 Treg cells (**h**). Source data are provided as a Source Data file.

suppressive activity[65] in favor of a subsequent pregnancy. In order to verify if these markers might by suitable to identify CD4+ mTreg cells, further confirmation is needed in additional in-vivo studies.

Both methods also identified CD20 as potential marker to distinguish between naïve and memory CD4+ Treg cells, although it is known as a classical B cell marker expressed by the majority of B cells[66]. However, recent reports also indicate that a small percentage of CD3+ T cells dimly express CD20 on their surface in human[67] and in mice[68]. First studies identified this T cell subset as highly inflammatory

implicated in autoimmunity, but they were also predominantly found to exhibit a memory phenotype[67]. Hence, the role and function of CD20+ T cells need further classification, not only in the context of pregnancy.

Additionally, our bioinformatic analyses further revealed CD73 and FR4 as potential markers to distinguish between naïve and mCD4+ Treg cells. However, due to their ubiquitous expression on CD4+ Treg cells in general, these markers may not be the most suitable ones to identify memory CD4+ Treg cells, but could still account for the

improved outcome in subsequent pregnancy. Together with CD39, CD73 enables CD4[+] Treg cells to convert ATP to immunosuppressive adenosine. Further, the expression of CD39 and CD73 on the surface of CD4[+] T cells and CD4[+] Treg cells is negatively correlated with the number of Th17 cells which could affect T cell plasticity[69]. CD73 expression might also be involved in the transdifferentiation of Th17 cells to CD4[+] Treg cells, hereby supporting fetal tolerance and fetal development in subsequent pregnancies. FR4 is important for the uptake of serum folates by cells expressing this receptor. It is highly expressed on natural CD4[+] Treg cells and plays a predominant role in the maintenance of the Treg phenotype[70,71]. Interestingly, CD73 and FR4 are both reported to be expressed on anergic T cells, which might help to sustain tolerance by expanding the CD4[+] Foxp3[+] Treg cell repertoire[72].

We here introduce the transdifferentiation of Th17 cells to CD4[+] Treg cells as a potential independent mechanism to improve maternal immune adaptation and pregnancy outcome in subsequent pregnancies. The uterus is a unique organ that is characterized by profound tissue remodeling, local inflammation due to infection and phases of substantial inflammation during pregnancy[53,73]. Hence, the balance of inflammatory and anti-inflammatory events could be maintained by the transdifferentiation of Th17 cells to CD4[+] Treg. However, this Th17/Treg balance, especially during pregnancy, remains poorly understood[74,75] and further studies are necessary to unveil the underlying mechanisms.

It is firmly established that the generation of CD4[+] Treg cells during pregnancy is driven by progesterone via DC-dependent pathways, and also glucocorticoids exert anti-inflammatory functions[37,76,77]. However, a hormonal impact on elevated CD4[+] Treg cells during second pregnancy could largely be excluded. We observed an enhanced frequency of tolerogenic DCs[78,79] in our present study, which may account for the increased generation or differentiation of CD4[+] Treg cells in second pregnancy. Vice versa, CD4[+] Treg cells can also condition DCs towards a tolerogenic phenotype by specifically down regulate the expression of CD80 and CD86 on DCs[80], which could also explain our present observations. Hence, the cross talk between DCs and CD4[+] Treg cells and their accountability for improved maternal immune adaptation needs further clarification.

Human studies point to an increased birth weight in subsequent pregnancies without a difference in gestational age at delivery when the first pregnancy went without complications[19]. In the present study, differences in fetal weight in response to a prenatal challenge were often marginal. However, it must be taken into consideration that various other physiologically occurring adaptational processes that are not compromised by antigen-specificity and immune-related specifics may also facilitate a second pregnancy's progression. This includes cardiovascular and haemodynamic changes such as cardiac output[81,82], vascular resistance[83] and also mean arterial pressure[84]. Further, adaptations occur in the renal systems, including an enhanced glomerular filtration[85], the pulmonary system[86] and the metabolic system to meet the nutritional demands of the fetus[87,88]. Finally, also other immune cell population such as CD8[+] T cells including CD8[+] CD122[+] regulatory T cells[40] and also B cells[89] will most likely contribute to gestational memory and improved maternal immune adaptation during subsequent pregnancies. Hence, further study assessing the impact of these cell populations will be crucial to comprehensively understand the mechanism of gestational immune memory.

In summary, in accordance with evidence arising from human studies supporting the presence of an immune memory during second pregnancies[19], our study advocates the generation of CD4[+] mTreg cells during a first uncomplicated pregnancy, which is paternal antigen-specific and rapidly expands during subsequent pregnancies. Hereby, immune tolerance towards the fetus is facilitated and fetal growth and development is improved. Our findings significantly improve our understanding of pregnancy-induced immune memory and hold the potential to foster the identification of immune targets aiming to reduce the risk for immune-mediated pregnancy complication.

## Methods
### PRINCE Study
The PRINCE (Prenatal Identification of Children's Health) study is conducted at the University Medical Center Hamburg-Eppendorf (UKE) and was initiated in 2011. Inclusion criteria were maternal age of 18 years or higher and a viable singleton pregnancy at gestational week 12–14. The following exclusion criteria were defined: women with chronic infections (HIV, hepatitis B/C), known substance abuse and smoking, multiple pregnancies or pregnancies conceived after assisted reproductive technologies. Pregnant women were invited to three antenatal visits, once per trimester (gestational weeks 12 to 14, 24 to 26, and 34 to 36). Data on the assessment of relevant covariables is described in detail elsewhere[90]. All study participants signed informed consent forms and the study protocol was approved by the ethics committee of the Hamburg Chamber of Physicians (PV3694). At the time of analyses, the PRINCE study sample consisted of 761 women. In the present analysis, data on women with two consecutive pregnancies were included ($n = 78$). For flow cytometry analyses, PBMCs obtained from 15 women from all three trimesters in first and second pregnancy were used.

### Study design for animal experiments
General experimental setup: Timed pregnancies was used in all experiments. Eight to ten-week-old female mice were allogenically mated overnight to Balb/c or OVA-Balb/c male mice, respectively. Mice underwent an undisturbed first pregnancy and were allowed to give birth. Litters born within the first pregnancy were removed within 24 h after giving birth to circumvent the hormonal changes associated with lactation and to quickly restore maternal hormonal levels to non-pregnancy levels. After 14 days, female mice were mated again to the same male. Aged-matched virgin mice and female mice that were mated just once served as controls, respectively. The presence of a vaginal plug in the morning was considered as gd 0.5. Maternal weight was controlled on gd 8.5 and 10.5 to confirm ongoing pregnancy. Experimental readout was done postpartum (Fig. 2) or at various days during pregnancy (gd. 3.5, 7.5 and 15.5).

Pregnancy outcome was assessed on the various experimental days and dependent of the gestational time point at which dams have been euthanised by cervical dislocation after $CO_2/O_2$ anesthesia. We documented the total number of implantations and the frequency of fetal loss (intrauterine hemorrhagic spots) per pregnant female. The abortion rate was calculated by the following equation: (number of intrauterine hemorrhagic spots /number of implantations) * 100. Additionally, fetal weight was assessed.

For CD4[+] Treg depletion, female 'depletion of regulatory T cell' (DEREG) mice were intraperitoneally injected with diphtheria toxin (DT, 500 µg/kg body weight) within 24 h after birth on two consecutive days. Hereby, all endogenous CD4[+] Treg cells were eliminated including CD4[+] Treg cells with a presumed memory phenotype that had been generated during the first pregnancy. Within eleven days after the last DT injection, normal CD4[+] Treg frequencies could be determined in the uterus-draining lymph node and the uterus. Hence, mice were subjected to a recovery phase of 14 days to restore normal CD4[+] Treg levels and subsequently re-mated with the same male for their second pregnancy.

For test for antigen-specificity, previously Balb/c mated mice were mated to DBA males for their second pregnancy. Further, we adoptively transferred 150.000 CD4[+] Treg cells from either virgin or parous mice intravenously injected to OVA-Balb/c mated females on gd 0.5, respectively. These CD4[+] Treg cells were harvested from uterus-draining lymph nodes and the uterus of virgin Fir/Tiger mice and Fir/Tiger mice previously mated twice to OVA-Balb/c males within 2 weeks

postpartum, and isolated and sorted by Fluorescence Activated Cell Sorting (FACS) under sterile conditions. The transfer of CD4[+] Treg cells obtained from virgin mice was necessary to control for the number of transferred CD4[+] Treg cells. Due to compliance with the 3 R principle, only a single injection of CD4[+] Treg cells was allowed.

## Mice

For the majority of experiments, we took advantage of a double knock-in reporter mouse line, the FoxP3-IRES-mRFP (Fir)/interleukin-10 (IL-10) IRES GFP-enhanced (Tiger) line, herein referred to as Fir/Tiger mice[44,91]. This mouse line, which is on a C57BL/6 background, allows the immediate and simultaneous detection of FoxP3 (red fluorescent) and IL-10 (green fluorescent) by flow cytometry without antibody labeling or other experimental procedures like permeabilizing the cell membrane. In order to specifically deplete CD4 Treg cells, we employed the DEpletion of REGulatory T cells (DEREG) mouse model. These mice are also on a C57BL/6 background and expresses a diphtheria toxin (DT) receptor–enhanced green fluorescent protein (eGFP) fusion protein under the control of the Foxp3 gene locus, allowing selective and efficient depletion of Foxp3[+] regulatory T cells by DT injection[92,93]. In order to assess CD4 Treg plasticity, we used the Fate mouse model, a combination of a IL-17A fate reporter mouse (IL-17A[CRE] × Rosa26 STOP[flox/flox] YFP (R26[YFP]))[43] with a IL-17A[Katushka] IL-10[eGFP] Foxp3[RFP] triple reporter mouse model[42,44,45]. Cells that have previously expressed high level of *Il17a* are permanently marked by YFP expression due to deletion of the stop cassette preceding R26[YFP]. Thus, former TH17 cells positive for YFP can be investigated for present IL-17A, IL-10 and Foxp3 expression ex vivo without in vitro restimulation. Adoptive transfer experiments were performed with wildtype C57Bl/6 J mice. All animals used in this study were single-housed (males) or maintained in groups (females) in the animal facility of University Medical Center Hamburg-Eppendorf with and kept under 12 h light/dark cycles at a room temperature of 21 °C and humidity controlled at 43%. Food and water were provided ad libitum. During experiments all involved animals were housed in the same animal barrier. Animals' care and all experimental procedures were performed in accordance with institutional guidelines and conform to requirements of the German Animal Welfare Act. Further, all experiments were carried out in accordance with the animal ethics approval given by the State Authority of Hamburg (G16/012, G17/049, N20/14 and ORG_1009).

## Tissue harvesting

Mice were anesthetized with $CO_2/O_2$ 4 weeks after last delivery or on various days throughout gestation, respectively. A blood sample was collected by retro bulbar puncture and subsequently mice were euthanised by cervical dislocation. The uterus-draining lymph node on both sides of the aorta was harvested and kept in PBS on ice. The fetuses and corresponding placentas were isolated from the amniotic membranes and the uterus was retained in HBSS on ice. Placentas were either stored at -20 °C in RNAlater (Ambion by Life Technologies GmbH) for subsequent gene expression analysis or embedded in biopsy cassettes and stored in 4% Formaldehyde solution (Sigma-Aldrich, St, Louis, US) for 24 h before transfer into 1% Formaldehyde solution for long-term storage and histological staining.

## Tissue processing and cell staining procedure

Single cell suspensions of maternal lymph nodes and uteri were obtained as follows[37,77]: Lymph nodes were mechanically disrupted and passed through a cell strainer. The uterus was enzymatically digested using 200 U/mL hyaluronidase (Sigma-Aldrich), 1 mg/mL collagenase VIII type (Sigma-Aldrich), and 1 mg/mL bovine serum albumin fraction V (Sigma-Aldrich) dissolved in 5 mL HBSS. Solution was incubated twice for 20 minutes in a 37 °C water bath with agitation and intermediately and subsequently recovered and filtered through a cell strainer. Finally, lymph node and uterine cell suspensions were

centrifuged at 450 g for 8 minutes at 4 °C and obtained cell pellets were resuspended in PBS. PBMCs were isolated from blood samples by using 1x Red Blood Cell (RBC) lysis buffer (eBioscience, Invitrogen by Thermo Fisher Scientific) according to the manufacturer's instructions. Lysis was stopped with PBS and subsequently, samples were centrifuged at $450 \times g$ for 8 minutes at 4 °C and resuspended PBS. Viable leukocyte count was obtained by using a Neubauer chamber upon adding Trypan Blue stain (0.4%, Life Technologies GmbH, Darmstadt, Germany).

For flow cytometric analyses, $1.0 \times 10^6$ maternal lymph node and uterine cells were used for immunophenotyping. Non-specific binding was blocked by rat anti-mouse CD16/CD32 Mouse Fragment crystallizable (Fc) Block (1:200, BD Bioscience) and Normal Rat Serum (1:100, eBioscience) for 15 min at 4 °C. Subsequently, cells were incubated with the respective antibodies for 30 min (Supplementary Table. 1). In order to identify dead cells, cells were simultaneously stained with eFluor 506 viability dye (eBioscience). For intracellular staining, cells were fixed and permeabilized using Foxp3 Fixation/Permeabilization Concentrate and Diluent (eBioscience) according to the manufacturer's instructions.

## Hormonal assessments

Maternal blood samples were centrifuged at 10.000 g for 20 min at 4 °C and the supernatant plasma was immediately frozen at −20 °C. For hormone analysis, plasma samples were diluted 1:200 or 1:500 using ELISA Buffer and measured with competitive immunoassays (Progesterone ELISA Kit, Cayman Chemical, Michigan, USA and Corticosterone ELISA Kit, Arbor Assays) and on a NanoQuant (Tecan Group AG, Männedorf, Switzerland) according to manufacturer's instructions.

## Placental histology

Paraffin embedded placentas were cut into 4 µm thick histological sections at the mid-sagittal plane using a microtome (SM2010R, Leica, Bensheim, Germany). Slides were deparaffinized and dehydrated twice using xylene and ethanol (70%). Masson-Goldner Trichrome Staining Kit (VWR International) was used to visualize the morphologically different areas of placental tissue. Hereby, tissue sections were stepwise stained with Weigert's iron hematoxylin, azophloxine staining solution, phosphotungstic acid orange G, and light-green SF solution following the manufacturer's instructions[40]. Subsequently, slides were scanned with a Mirax Midi Slide Scanner (Zeiss). Histomorphological analyses of placental area (junctional zone and labyrinth) were performed by two independent observers using Panoramic Viewer (3DHistech Kft. Budapest, Hungary).

## Confocal microscopy

The gd 7.5 implantations were removed from the uterus in PBS on ice and transferred to a fresh dish. The decidua was removed and the trophoblast was peeled away, followed by fixation with 4% paraformaldehyde for 30 minutes. Afterwards, the embryos were washed with PBS three times. For permeabilization embryos were put in a solution of 0.5% triton in PBS. Subsequently, the embryos were washed in PBS-tween (0.1% tween 20 in PBS) and 4 mg/ml BSA (PBS-T/BSA) three times for one hour. After fixation and permeabilization, the embryos were stained and incubated with the SRY (sex determining region Y)-box 2 (SOX2, 1:100) and NANOG (1:200) antibody cocktail diluted in PBS-T/BSA at 4 °C overnight. The following day, the embryos were washed in PBS-T/BSA three times, followed by DAPI (1:1000 in PBS-T/BSA) staining for 15 minutes. The embryos were mounted onto chamber dishes in Citiflour and confocal microscopy was performed immediately. The optical slices were 5−7.5 µm thick and an approximate depth of 250 µm could be achieved for most of the embryos, which is about half way through the embryo. The embryos were sequentially scanned in different channels. The images were analyzed using the Imaris software.

## Infinity Flow and direct cell set analysis of parallel flow cytometry data

Uterus-draining lymph node from virgin ($n = 6$) and parous ($n = 3$) Fir/Tiger mice were isolated and subsequently stained for CD45.2 (one fluorochrome per specimen), CD3, CD4, CD8 and CD25. Afterwards, both samples were pooled and subjected to the LEGENDScreen™ Mouse PE Kit (Biolegend) containing 255 PE-conjugated monoclonal antibodies against various mouse cell surface markers, plus 11 mouse, rat, and hamster Ig isotype controls, arrayed on three 96-well plates. Plates were acquired using a BD LSRFortessa II with a high throughput sampler (HTS). After FCS file export from the flow cytometer, files were cleaned, post-compensated and pre-gated to CD3. Using the R package InfinityFlow (v1.6.0) with XGBoost modelsthe values of exploratory markers were imputed and UMAPs were generated[46]. The average expression of each marker across the two conditions was calculated. Subsequently, the most discriminating markers were identified by computing the Area Under the ROC Curve (AUC). The AUC was calculated for each marker and represents a marker's ability to distinguish cells from virgin or parous mice. The closer to 0 or 1, respectively, the better at separating the two conditions, while an AUC close to 0.5 corresponds to a random guess.

Infinity Flow is a machine learning based approach that allows for higher cell counts and thus may yield discriminating markers that can otherwise not be identified. To complement Infinity Flow, an additional direct analysis of each marker on the corresponding cell subset was conducted. Here, for an interpretable selection of the most discriminating markers, the effect size of virgin vs. parous for each marker was calculated using Cohen's d and the median fold change was computed. A Cohen's $d$ value of 0.2 is considered a small effect[94]. Markers with an absolute effect size of more than 0.3 were considered as discriminating candidates to be further filtered via median fold change. Additionally, the Wilcoxon Rank Sums test was applied to account for small cell counts, i.e., only markers with an associated $P$ value of $> 0.05$ (after Benjamini-Hochberg correction) were considered.

### Statistical analyses

All statistical analysis was performed using GraphPad Prism 9 (GraphPad Software, San Diego, CA, USA). All data obtained from each experiment are presented as violin plots with individual point, median and quartiles. Data were checked for outliers identified by the ROUT method offered within GraphPad Prism. The statistical significance between two groups were assessed using Student's t-test. The statistical significance between more than two groups was calculated using One-way-Anova with Bonferroni test to correct for multiple comparisons (* $p < 0.05$, ** $p < 0.01$, *** $p < 0.001$).

### Reporting summary

Further information on research design is available in the Nature Portfolio Reporting Summary linked to this article.

## Data availability

All data presented in this study are available in the article and its source data, or from the corresponding authors upon request. The flow cytometry data generated in this study have been deposited to the research data repository of the University of Hamburg (www.fdr.uni-hamburg.de) under the following DOIs: human data: 10.25592/uhhfdm.17577 and mouse data: 10.25592/uhhfdm.17579. Additional data can be requested from the corresponding authors. Source data are provided with this paper.

## Code availability

Infinity Flow is implemented as an R package, infinityFlow (https://www.github.com/ebecht/infinityFlow).

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

## Acknowledgements

The authors would like to thank Agnes Wieczorek, Thomas Andreas (Department of Obstetrics and Fetal Medicine, University Medical Center Hamburg) and Danny Schreier for their technical assistance and Björn von Schlippe for graphical support. The flow cytometer and sorter used in this study were provided by the FACS Core Unit of the University Medical Center Hamburg-Eppendorf. We thank the NIH Tetramer Core Facility (contract number 75N93020D00005) for providing (PE-labeled - I-Ad chicken ova 323-339 ISQAVHAAHAEI-NEAGR) tetramers. We thank for the financial support by grants from the German Research Foundation (Deutsche Forschungsgemeinschaft, DFG) to KT (TH 2126/1-1) and PCA (AR232/26-2, AR232/27-2, AR232/29-2), from the Authority for Science, Research and Equality, Hanseatic City of Hamburg to PCA (LFF-FV73), from the Excellence Strategy of the Federal Government and the Länder to PCA (Z5-V2-008: TN 2023 Preterm) and from the Federal Ministry of Education and Research (Bundesministerium für Bildung und Forschung, BMBF) as part of the German Center for Child and Adolescent Health (DZKJ) to PCA and AD (funding code 01GL2404A). LSA and RW were supported by the Else Kröner-Fresenius-Stiftung iPRIME Scholarship and JIAR was supported by a Scholarship provided by the IRTG of the CRC1192 (DFG). We further like to acknowledge the financial support from the Open Access Publication Fund of UKE - Universitätsklinikum Hamburg-Eppendorf.

## Author contributions

Conceptualization: K.T. and P.C.A.; Methodology: K.T., P.C.A., E.B. and M.B.; Investigation: K.T., C.U., J.I.A.R., L.S.A., R.W., S.S., J.J.E., E.B., M.B. and M.Q.; Recruitment of PRINCE study participants: A.D.; Coordination and Funding of the PRINCE study: A.D. and P.C.A.; Visualization and presentation of data: K.T.; Project funding acquisition: K.T. and P.C.A., Project administration: K.T., Supervision—overall project: P.C.A.; Supervision—individual parts of the project: N.G., H.W.M., M.A., Writing —original draft: K.T., P.C.A., Writing—review & editing: all authors.

## Funding

## Competing interests

The authors declare no competing interests.
