## [Transparent Peer Review file · Nature Communications]

Pregnancy-acquired memory CD4+ regulatory T cells improve pregnancy outcome in mice

Corresponding Author: Dr Kristin Thiele

Version 0:

Reviewer comments:

Reviewer #1

(Remarks to the Author)

This manuscript investigated the role of pregnancy-acquired memory CD4+ regulatory T cells (CD4+ mTregs) in improving pregnancy outcomes, particularly in subsequent pregnancies in mice. The study addresses an important and novel question in reproductive immunology, with a robust experimental design.

Major Issues:

1. The use of DT to deplete Tregs in this study raises concerns about potential non-specific effects on other cell types, which could indirectly influence pregnancy outcomes and complicate result interpretation. Furthermore, the manuscript lacks sufficient control experiments to account for the direct effects of DT alone on Tregs and pregnancy outcomes. The possibility that DT could affect the maternal immune system in early pregnancy, influencing outcomes, is also raised. To address these concerns, it is suggested to include control groups that specifically evaluate the effects of DT alone, which would help distinguish Treg depletion effects from those of DT.
2. While the study is well-designed, its novelty is somewhat limited, as Treg depletion and its effects on pregnancy outcomes have been extensively studied (DOI: 10.1038/nature11462; DOI: 10.1126/science.adf9325). The use of DERE mice is well-established in such investigations. To enhance the manuscript's uniqueness, it is recommended that the authors clearly outline the innovative aspects of this study, focusing on the specific new questions it addresses that differentiate it from previous work.
3. As the number of pregnancies increases, the proportion of fetal antigen-specific immune cells, such as CD8+ T cells (which could be cytotoxic), may also rise. It would be beneficial to explore whether there is an antagonistic/synergistic relationship between fetal antigen-specific Tregs and other immune cells, particularly CD8+ T cells and B cells, and how Tregs regulate their activity to maintain immune tolerance. Additionally, further investigation into changes in related inflammatory markers and adaptive immune indicators could provide evidence of Treg's role in sustaining immune tolerance.
4. Did the authors examine Tregs in different tissues, such as mesenteric lymph nodes, peripheral blood, or decidua? It would be valuable to know if Tregs expressing candidate markers are observed in these locations and if there are differences between groups. This could also help to determine whether Tregs increase specifically in the uterine-draining lymph nodes or are also detectable in easily accessible clinical samples, such as peripheral blood and decidua.
5. The data indicate that IL-10 may be involved in the function of fetal antigen-specific Tregs. Additional experiments, such as using IL-10 knockout Tregs or CD4+ T cells, are needed to determine whether IL-10 is a key effector molecule mediating maternal-fetal tolerance.

Minor Issues:

6. Line 40 Wording Revision: The sentence "higher and fetal-antigen-specific CD4+ T regulatory (Treg) cell frequencies..." should be revised for clarity: "We identified higher frequencies of fetal-antigen-specific CD4+ regulatory T (Treg) cells both postpartum and in subsequent pregnancies in mice."
7. Figure 1 (G, J, K): The flow cytometry plots and statistical graphs should display CD44 and FOXP3 on the x- and y-axes rather than showing CD44 and CD62L co-expression in lymphocyte subsets. This adjustment would better highlight the expression characteristics of the CD44^{high} Treg subset and the differences in its expression between groups.
8. Figure 2: It is recommended to include an additional time point before pregnancy (gd0.5) to monitor changes in Treg populations at this stage.
9. Figure 3: The data measured in Figure 3 should be combined with the data from Figure 1, as they support similar conclusions and would streamline the presentation.
10. Figure 4: The tetramer detection method does not specify the type of antibody used. This information should be added to

the experimental methods section.

11. Figure 5 Legend: The explanation for "DF = different father" should be moved from Figure 5 to the legend of Figure 4 for better clarity.

Reviewer #2

(Remarks to the Author)

In the paper "Pregnancy-acquired memory CD4⁺ regulatory T cells improve pregnancy outcome in mice" Thiele A. et al show that, during the first pregnancy there is an initiation of immune memory characterized by the expansion of memory CD4⁺ Treg (CD4⁺ mTreg) cells in mice, linked with an improved fetal outcome. By using a high-throughput single cell quantification method it was possible to identify CD4⁺ mTreg cells during gestation and also in other settings. Despite this study provides a basis for elucidating the function of CD4⁺ Treg cells in the context of pregnancy outcomes, several issues should be clarified to improve the study. Mechanistically, one of the major limitations of the study is that most of the data reported are obtained in mouse models, with no attempt to suggest/show translation to humans.

Major comments for revisions:

- 1) The interaction between the immune system and the endocrine system is known to contribute to creating a favorable environment for the fetus. Since several hormones associated with pregnancy, such as hCG and steroid hormones, (as estrogens, FSH and LH), participate in the maintenance of immune homeostasis, the authors should evaluate hormonal asset in pregnant mice with higher and lower number of D4⁺ mTreg cells.
- 2) It would be useful for the authors to investigate which molecular targets/signatures possess these CD4⁺ mTreg cells, whether they have a specific gene signature that identify a specific molecular asset fro pregnancy-related mTreg.
- 3) To complete the analysis of the CD4⁺ mTreg compartment, the authors could assess the status of the periphery (spleens) to observe whether there is an increase in CD4⁺ mTreg cells in subsequent pregnancies. This would also allow them to assess immune cell levels at time zero, before any pregnancy, and subsequently.
- 4) The authors conducted a comprehensive analysis of the frequencies of CD4⁺ mTreg cells during the postpartum period and the second pregnancy in mice. However, to gain a deeper understanding of the role of these cells, it would be beneficial for the authors to perform a series of functional experiments to assess whether the increase in frequencies observed during the second and third pregnancies is associated with enhanced functionality of these cells. It would be useful for the authors to purify Treg cells and evaluate their suppressive capacity in vitro, using CD4⁺ cells from the same mouse from which the Treg cells were isolated and CD4⁺ cells from a non-pregnant mouse, in order to more accurately assess their functionality. Similarly, the authors could strengthen their findings of an increased level of regulatory T cells following a second and third pregnancy by analyzing several markers related to the suppressive function of these cells and assessing whether these Treg cells are stable over time.
- d) In order to gain a deeper insight into the inflammatory processes that are known to occur during pregnancy and that could potentially contribute to miscarriage, the authors should perform measurement of circulating cytokine levels to correlate with increased frequencies of Treg cells that they observed (IL10, TGFb, etc).

Minor revisions:

- a) To gain a deeper insight into the percentage of mTreg cells illustrated in Figure 1 G and K, the authors may wish to consider selecting a more representative flow cytometry plot and, most importantly, presenting the graphs as violin plots, with the individual percentage values obtained from the different mice.
- b) In Figure 2 I and K, it would be useful to include the values corresponding to gd 3.5, if they are too low, the authors might consider splitting the y-axis in this case.
- c) In Figure 2E, the authors show the percentage of CD4⁺ FoxP3⁺ cells (Treg cells). The violin plot shows very low values, the highest value is about 1%. To be consistent and maintain the same style as the other figures, it would be useful to show a cytofluorimetry plot representing these percentages.

Reviewer #3

(Remarks to the Author)

In this study, the authors investigated memory T cell function in pregnancy. Using a high-throughput single-cell quantification method, authors found candidate markers such as CXCR4 and CD274 for detecting CD4⁺ mTreg cells. It is reported that findings may contribute to the improved understanding of pregnancy-induced immune memory and foster the identification of immune targets aiming to reduce the risk for immune-mediated pregnancy complications.

The study is well-designed to explain the fundamental questions for the mTreg cells in pregnancy using a murine model. This work is a significant addition to the field of reproductive immunology and obstetrics and gynecology. The work supports the conclusion and claims. The methodology is sound, and there is a clear explanation for the limitations of the study. The methodology is well written for the work to be reproduced.

The following are comments and concerns.

Figure 1.

The increased mTregs are limited to the draining LN but not in the uterus: The frequencies of CD44⁺⁺ /CD4 Treg remained

unaffected in the uterus with the increasing number of pregnancies. Please explain this finding further in the discussion. The title of Figure 1 can be "Improved neonatal outcome and mTreg and Treg cells in subsequent pregnancies."

Are there any age-related changes in the Treg cell population since the controls were virgin mice since the 2nd and 3rd pregnancies were at an advanced age compared to virgin mice?

The litter size and neonatal weight were increased with the increasing orders of pregnancies. However, in LN, there were no differences regarding Treg cells between 2nd and 3rd pregnancies, although there seems to be a stepwise increase in mTreg cells. What is a possible explanation for this finding? If the litter size, neonatal weight, and Treg cells keep increasing with the advanced number of pregnancies, it can be related to other concerns. Please discuss.

Figure 2.

*Please mark Figure 2, D as LN and Figure 2, G as uterus.

*In the Figure 2 legend, please delete the space in H. I,J. Two-way-Anova should be Two-way ANOVA. No DC in 3.5 weeks of gestation during 1 and 2nd pregnancy? Please consistently number the figures: Authors describe "Fig 2B-D" or "Fig 3, H-J."

*In Figure 2B, C, E, F, H, I and K, please add a legend to explain yellow and green violin plots

Serum progesterone level goes up with the advance of pregnancies. In this study, serum progesterone level was decreased at 7.5 compared to 3.5 gd.

Figure 3. Please specify that Sox2 was significantly increased but not for Nanog (line 144). Again, labyrinth changes are not statistically significant.

Is Figure 3D a staining of the whole fetus? Please explain the staining outcomes in detail.

*In Figure 3 J, green and blue lines are hard to see. It is recommended to separate the tracing of these lines without background histopathologies. The tracing can be added to each placental cut section.

*In Figure 4, please add the explanation for AT. In the legend, 150,000 CD4+ Treg should be 150,000 CD4+ Treg. 4J and L. What does P stand for in J, L, N, and P?

**"student t-test" should be Student t-test.

Figure 4.

nTreg was adoptively transferred, uterine or LN Tregs did not increase. Is there any possibility that allo rejection may play a role in this finding?

Figure 5.

Figure 5 B. Was the abortion rate 0% for not depleted animals?

Line 239. Prenatal stress was given on gd 10.5, 12.5 and 14.5. In the graph, it is listed as gd 15.5.

Line 225. The authors stated that the abortion rate was not changed as expected due to the late stress exposure during gestation. What is the abortion rate when the stress was given on gd 10.5 or 12.5? In the figure legend, it seems that experiments were also performed on those days.

Figure 6.

In 2nd pregnancy, the % ExIL17 and Tr1 cell proportion were significantly increased compared to the first pregnancy. It is postulated that these changes may be induced upon implantation or labor-related inflammation during the first pregnancy. Th17 cells increase during these conditions. Is there any other evidence that increased Th17 cells induce transdifferentiation to Treg cells?

*Figure 6B. Please add legends for different color violin plots.

Figure 7

*In Fig 7 G, Please add the MFI legend displayed.

Discussion

A lack of TRM cells in the uterus was explained as a lack of proper markers. The difference between the uterus and other organs, such as the skin or small intestine, is menstruating features. While the functional layer of the endometrium is exfoliating,

Where is the actual location of TRM cells in the endometrium or decidua spacially?

Line 353-360: Endometrial TRM cells also express CD49a, CCR5, and PD1. CCR5 expression was related to IL17 production. Recently, tissue-resident memory T cells have been reported in the murine uterus. This section can be enhanced by updating the reported data regarding TRM cells in the uterus.

Lines 421-430: The possible impact of over-expansion of Treg and mTreg cells should be considered regarding the advanced number of pregnancies.

Version 1:

Reviewer comments:

Reviewer #1

(Remarks to the Author)

The authors have provided comprehensive clarifications in rebuttal to my prior comments. This manuscript will be a nice contribution to the field.

Reviewer #2

(Remarks to the Author)

Authors have attempted to reply to the majority of my comments and the paper overall has been improved.

Reviewer #3

(Remarks to the Author)

Review of the Authors' Response to Reviewer Comments

The authors have provided a thorough and well-structured response to the reviewers' comments. Most concerns and suggestions have been adequately addressed with appropriate revisions and additional experiments. Below is an assessment of their responses:

General Improvements:

The authors have satisfactorily addressed all minor errors and concerns.

Additional experiments have been conducted to evaluate the potential detrimental effects of DT, and the results appear to support their conclusions.

The uniqueness of the study has been clarified, strengthening its contribution to the field.

The revised discussion adequately addresses the immune cell populations involved in gestational immune memory.

Data for PBMC and other tissues have been included, showing no significant changes, which strengthens the conclusions.

The justification for not including IL-10 KO mice experiments is reasonable.

Flow cytometric plots have been updated to display CD44 and FOXP3, improving clarity.

The authors have provided a valid explanation for not including gd0.5 data.

Specific Concerns:

The lack of a translational approach has been addressed with the addition of human data, which improves the study's relevance.

The response regarding Figure 1E is noted. While the authors reported improved intrauterine growth in the second and third trimesters of subsequent pregnancies, data from the first pregnancy are missing from the plot. It would be helpful for the authors to clarify whether these data are available and, if not, provide further justification.

The interaction between the immune and endocrine systems was further investigated, and the conclusion that beneficial effects are primarily cell-mediated rather than hormone-dependent is well-supported.

The rationale for not including molecular targets/signatures for CD4+ mTreg cells is acceptable.

Concerns about age-related changes have been clarified with additional explanations.

Amendments regarding Sox2 and Nanog adequately address previous concerns.

The issue of possible Th17-to-Treg transdifferentiation has been sufficiently addressed.

Final Assessment:

Overall, the authors have provided a comprehensive and well-reasoned rebuttal. Most concerns have been resolved with appropriate data additions or clarifications. The only remaining issue pertains to the absence of first-pregnancy data in Figure 1E, which should be explicitly clarified in the revised manuscript. Aside from this minor point, the response is satisfactory, and the manuscript is much improved and ready for publication after resolving a minor issue.

Sincerely,

Joanne Kwak-Kim

Point-by-point reply to reviewers' comments on manuscript entitled 'Pregnancy-acquired memory CD4⁺ regulatory T cells improve pregnancy outcome in mice' by K. Thiele *et al.* (NCOMMS-24-48524)

Reviewer comments:

Reviewer #1:

General comment from this reviewer: This manuscript investigated the role of pregnancy-acquired memory CD4⁺ regulatory T cells (CD4⁺ mTregs) in improving pregnancy outcomes, particularly in subsequent pregnancies in mice. The study addresses an important and novel question in reproductive immunology, with a robust experimental design.

Authors' response: We thank the reviewer for this initial assessment of our study and would like to comment on the following specific issues raised by the reviewer.

Reviewer's comment #1: The use of DT to deplete Tregs in this study raises concerns about potential non-specific effects on other cell types, which could indirectly influence pregnancy outcomes and complicate result interpretation. Furthermore, the manuscript lacks sufficient control experiments to account for the direct effects of DT alone on Tregs and pregnancy outcomes. The possibility that DT could affect the maternal immune system in early pregnancy, influencing outcomes, is also raised. To address these concerns, it is suggested to include control groups that specifically evaluate the effects of DT alone, which would help distinguish Treg depletion effects from those of DT.

Authors' response: We thank the reviewer for raising this point. We have carefully established the protocol to deplete Tregs using the DEREK mouse model, which included control for potential direct diphtheria toxin (DT) effects. To address this, we used age-matched C57Bl/6J females instead of DEREK mice and allogeneically mated these to Balb/c males once or twice, respectively. Subsequently, some second pregnancy mice were injected with DT after the first delivery and subjected to a recovery phase of 14 days before re-mating to match the experimental set-up used for the assessment of Treg depletion in DEREK females. On gestational day (gd) 15.5, abortion rate, fetal weight as well as CD4⁺ Treg cell frequencies in lymph node and uterus were investigated in these C57Bl/6J females. We could exclude direct DT effects on abortion rate and fetal growth. Additionally, as expected, no decline in CD4⁺ Treg frequencies upon DT injection in lymph node and uterus was detectable in these C57Bl/6J control females (see new Fig. S2C).

In addition to the exclusion of direct DT effects, we had performed additional control experiments, e.g., dose-response assessments to identify the optimum dosage sufficiently depleting all CD4⁺ Tregs in various organs and the recovery time to ensure that naïve CD4⁺ Tregs are present at the time point of second mating. Please see new Fig. S2: We used two different dosages (250 µg/kg and 500 µg/kg bodyweight) as a single dose, but that did not sufficiently deplete all CD4⁺ Tregs in the lymph node (A, top), although it was sufficient for the peripheral blood. Hence, we decided to use two consecutive injection at an interval of 24 hours (A, bottom). Subsequently, we monitored the CD4⁺ Treg recovery (Fig. S2B).

We amended the result section as follows:

“We established a dose of diphtheria toxin (DT) for sufficient CD4⁺ Treg cell depletion during the interpregnancy interval (Fig. S2A), Further, we monitored the recovery of the CD4⁺ Treg cells to ensure normal CD4⁺ Treg cell frequencies before the second mating (Fig. S2B). Additionally, to exclude side effects of DT on our analysed parameters, we performed a control experiment using C57B/6J mice and did not observed any alterations of fetal outcome and Treg frequencies (Fig. S2C)”

Fig. S2: Establishment of the DEREg (DEpletion of REGulatory T cells) mouse model:
(A) DEREg mice were injected with 250 or 500 $\mu\text{g}/\text{kg}$ bodyweight diphtheria toxin (DT), either once (top) or twice at an interval of 24 h (bottom). CD4⁺ Treg cell frequencies in the lymph node were determined 24 h after the last DT injection.
(B) Mice were injected twice with 500 $\mu\text{g}/\text{kg}$ bodyweight DT and CD4⁺ Treg cell frequencies in the lymph node were assessed on different days post-injection (dpi).
(C) In order to assess side effects of the DT injection, age-matched C57Bl/6J mice were allogeneically mated to Balb/c males once or twice, respectively. Subsequently, some second pregnancy mice were injected with diphtheria toxin (DT) after the first delivery and subjected to a recovery phase of 14 days before re-mating. On gestational day (gd) 15.5, abortion rate, fetal weight as well as CD4⁺ Treg cell frequencies in lymph node and uterus were investigated. Data are presented as violin plots with individual points, median and quartiles, and the statistical significance between groups was calculated using One-way-Anova (* $p < 0.05$).

Reviewer's comment #2: While the study is well-designed, its novelty is somewhat limited, as Treg depletion and its effects on pregnancy outcomes have been extensively studied (DOI: 10.1038/nature11462; DOI: 10.1126/science.adf9325). The use of DEREg mice is well-established in such investigations. To enhance the manuscript's uniqueness, it is recommended that the authors clearly outline the innovative aspects of this study, focusing on the specific new questions it addresses that differentiate it from previous work.

Authors' response: We thank the reviewer for this comment. We are well aware of these two publications as the first one is actually the one inspiring our study presented here. However, in both publications, a mouse model with supra-physiological class II paternal MHC expression on fetal tissues was used, along with the overexpression of the corresponding T cell receptor in the mother. As this may promote an overshooting Teff response¹, an evaluation of findings under conditions that can be considered as more physiological models are needed to understand the mechanisms underlying CD4⁺ Treg trajectories, expansion pathways and acquisition of memory markers in subsequent pregnancies. Further, the articles quoted by the reviewer only marginally touch the consequences for pregnancy outcome, e.g. fetal outcomes², or report artificially amplified fetal loss rates of 50-75%⁴, which exceed the generally reported fetal loss rate in C57Bl/6J mice. Clearly, this excess fetal loss can be explained by the use of the OVA:2W1S⁺ mouse model overexpressing a non-physiological class II paternal MHC. Yet,

validation of these finding in a more physiological setting using WT mice, as we provide with our study, is a sensible amendment to our understanding.

To address these aspects in the revised manuscript, we have amended the introduction as follows:

“First experimental evidence have been provided for the existence of memory CD4⁺ Treg (CD4⁺ mTreg) cells in mice, specifically into the antigen-specificity of these cells ² and the impact of fetal microchimeric cells for the re-expansion of CD4⁺ Treg cells ⁴. Since these evidences were based on a mouse model with an immune-dominant I-Ab:2W1S55–68 peptide expressed as a surrogate fetal antigen¹, confirmation under physiological settings in mice is needed. Further, the functional role of CD4⁺ mTreg cells, such as a re-expansion-related improved outcome in subsequent pregnancies is still elusive. The identification CD4⁺ mTreg cells is still ambiguous and often limited to CD45RO in human and CD44 in mice ^{6,8–10}.”

Reviewer’s comment #3: As the number of pregnancies increases, the proportion of fetal antigen-specific immune cells, such as CD8⁺ T cells (which could be cytotoxic), may also rise. It would be beneficial to explore whether there is an antagonistic/synergistic relationship between fetal antigen-specific Tregs and other immune cells, particularly CD8⁺ T cells and B cells, and how Tregs regulate their activity to maintain immune tolerance. Additionally, further investigation into changes in related inflammatory markers and adaptive immune indicators could provide evidence of Treg’s role in sustaining immune tolerance.

Authors’ response: We agree with the reviewer that it will be essential to investigate other immune cell population in respect to gestational immune memory. We already have preliminary data on CD8 T cells, here especially regulatory CD8 CD122 T cells might play a role due to their relevance for fetal growth and development in the later stages of pregnancy. Their frequency is also increased in subsequent pregnancy – please see below – suggesting a functional relevance to gestational immune memory. However, comprehensively addressing that would exceed the scope of the manuscript. Therefore, we have addressed this issue in the discussion:

“Finally, also other immune cell population such as CD8⁺ T cells including CD8⁺ CD122⁺ regulatory T cells (41) and also B cells (83) will most likely contribute to gestational memory and improved maternal immune adaptation during subsequent pregnancies. Hence, further study assessing the impact of these cell populations will be crucial to comprehensively understand the mechanism of gestational immune memory.”

Reviewer’s comment #4: Did the authors examine Tregs in different tissues, such as mesenteric lymph nodes, peripheral blood, or decidua? It would be valuable to know if Tregs expressing candidate markers are observed in these locations and if there are differences between groups. This could also help to determine whether Tregs increase specifically in the

uterine-draining lymph nodes or are also detectable in easily accessible clinical samples, such as peripheral blood and decidua.

Authors' response: We thank the reviewer for this question. We did further analyses and have now included additional results from tissues other than the paraaortic lymph node and the uterus, e.g. the inguinal lymph node as well as peripheral blood and spleen. While the inguinal lymph nodes behaved similar to their paraaortic counterpart, albeit less strong, we could not detect any differences in CD4 Treg frequencies in peripheral blood and spleen. We included an additional set of data regarding peripheral CD4 Treg frequencies in Fig. 1, B and C and Fig.S1 A-C to illustrate these findings.

Reviewer's comment #5: The data indicate that IL-10 may be involved in the function of fetal antigen-specific Tregs. Additional experiments, such as using IL-10 knockout Tregs or CD4⁺ T cells, are needed to determine whether IL-10 is a key effector molecule mediating maternal-fetal tolerance.

Authors' response: We thank the reviewer for this suggestion. However, IL-10 KO mice exhibit an altered lymphocyte and myeloid immune profiles and show altered responses to inflammatory or autoimmune stimuli. Further, these mice demonstrate an increased prevalence of colorectal adenocarcinoma and spontaneous development of chronic enterocolitis. Since pregnancy requires an adaptation to various organ systems, that may be influenced by the lack of IL-10, we are worried how that affects pregnancy itself and controlling for different confounding factors in 1st and 2nd pregnancy, respectively. A mouse line exhibiting an IL-10 KO on Tregs is not yet available to us. However, our results only convincingly show increased IL-10 in CD4⁺ Tregs before implantation (gd 3.5, Fig. 2 C and F), but increased frequencies of CD4⁺ Treg cells throughout pregnancy. Further, IL-10 was also elevated in DCs, also in later stages of pregnancy. Consequently, to comprehensively dissect the role of IL-10 in the context of gestational memory would require a complete set of experiments with different mouse models and experimental approaches that would exceed the content of this manuscript. Therefore, we have discussed this important aspect in the discussion.

“CD4⁺ Treg function are mediated by secretion of cytokines such as IL-10 and TGF- β . Specifically, IL-10 could play a special role here, since elevated IL-10 production in subsequent pregnancies was not only observed in CD4⁺ Treg cells, but also in DCs. Further, that increase was most pronounced in early pregnancy (gd 3.5) which might contribute to the almost complete absence of abortions in subsequent pregnancies as decreased levels of both, CD4⁺ Treg cells and IL-10, are linked to recurrent pregnancy loss (55). Hence, in further studies it will be important to dissect the impact of CD4⁺ Treg cells from the functional relevance of IL-10 as a potential key molecule by using e.g. IL-10 knockout mice or more specifically creating a conditional knockout of IL-10 on either CD4⁺ Treg cells or DCs.”

Reviewer's comment #6: Line 40 Wording Revision: The sentence "higher and fetal-antigen-specific CD4⁺ T regulatory (Treg) cell frequencies..." should be revised for clarity: "We identified higher frequencies of fetal-antigen-specific CD4⁺ regulatory T (Treg) cells both postpartum and in subsequent pregnancies in mice."

Authors' response: We thank the reviewer for this comment and have revised the sentence accordingly.

Reviewer's comment #7: Figure 1 (G, J, K): The flow cytometry plots and statistical graphs should display CD44 and FOXP3 on the x- and y-axes rather than showing CD44 and CD62L co-expression in lymphocyte subsets. This adjustment would better highlight the expression characteristics of the CD44^{high} Treg subset and the differences in its expression between groups.

Authors' response: We thank the reviewer for this suggestion. We have revised the figure accordingly.

Reviewer's comment #8: Figure 2: It is recommended to include an additional time point before pregnancy (gd0.5) to monitor changes in Treg populations at this stage.

Authors' response: We thank the reviewer for this recommendation. However, we have decided to refrain from adding data from figure 1 (virgin vs 1. interpregnancy interval) into figure 2 (B and E) as an additional time point. We agree with the reviewer that the pre-pregnancy time point important – that was the reason for figure 1. Due to experimental setup pursued in figure 1 (with three consecutive pregnancies), the age of the mice is quite different between figure 1 and 2 (approx. 21 weeks to 13 weeks). Further, the experiments were carried out at different times and although we always used the same flow cytometry panel, we cannot exclude variations due to technical issues on the flow cytometer. We have deliberately avoided using a solid line between the time points because, although we want to suggest a kinetics, the results at the individual time points were collected independently of each other. That means, we have prioritised the comparison of 1st versus 2nd pregnancy over the comparison between the individual time points.

Reviewer's comment #9: Figure 3: The data measured in Figure 3 should be combined with the data from Figure 1, as they support similar conclusions and would streamline the presentation.

Authors' response: We thank the reviewer for this suggestion. Due to including the flow cytometry data from the blood in Figure 1, we decided to remove the data regarding neonatal outcome (litter size and neonatal weight) and incorporated them into Figure 3, now illustrating pregnancy outcome data as a whole at birth and on gestation days 7.5 and 15.5. These data indeed support the same conclusion. The text in the manuscript has been revised accordingly.

Reviewer's comment #10: Figure 4: The tetramer detection method does not specify the type of antibody used. This information should be added to the experimental methods section.

Authors' response: We agree with the reviewer and apologize for this oversight. We have added the tetramer to Table 1 and amended the Acknowledgement section as follows:

"We thank the NIH Tetramer Core Facility (contract number 75N93020D00005) for providing (PE-labeled - I-Ad chicken ova 323-339 ISQAVHAAHAEINEAGR) tetramer."

Reviewer's comment #11: Figure 5 Legend: The explanation for "DF = different father" should be moved from Figure 5 to the legend of Figure 4 for better clarity.

Authors' response: We thank the reviewer for detecting this oversight. We have added the explanation to the figure legend of Fig. 4.

Reviewer #2:

General comment from this reviewer: In the paper "Pregnancy-acquired memory CD4+ regulatory T cells improve pregnancy outcome in mice" Thiele A. et al show that, during the first pregnancy there is an initiation of immune memory characterized by the expansion of memory CD4+ Treg (CD4+ mTreg) cells in mice, linked with an improved fetal outcome. By using a high-throughput single cell quantification method it was possible to identify CD4+ mTreg cells during gestation and also in other settings.

Despite this study provides a basis for elucidating the function of CD4+ Treg cells in the context of pregnancy outcomes, several issues should be clarified to improve the study. Mechanistically, one of the major limitations of the study is that most of the data reported are obtained in mouse models, with no attempt to suggest/show translation to humans.

Authors' response: We thank the reviewer for their summary of our study. Regarding the human translation: we agree with the reviewer that this is an important aspect. To address this, we have performed additional experiment using PBMC obtained from pregnant women in their 1st and 2nd pregnancies, respectively. Additionally, we can provide meta-data from 78 women with two consecutive pregnancies (new Figure 1). We observed a higher birth weight and a decline in pregnancy complications in 2nd pregnancy, albeit levels of significance were not reached, possible due to the n=78 the analysis may still be underpowered. Further, we did not

detect any differences in the PBMCs of pregnant women in their 1st and 2nd pregnancy. Hence, PBMC might not reflect the Treg situation at feto-maternal interface, which highlights the importance of our novel findings in mice. We have amended the result section as follows.

Less obstetric complications and an improved fetal growth in subsequent human pregnancies

Within the prospective longitudinal pregnancy cohort study PRINCE (Prenatal Identification of Children's Health), conducted at the University Medical Center Hamburg-Eppendorf, we have 78 women who participated with two consecutive pregnancies (Fig. 1A). Maternal age is naturally significantly higher in second compared to first pregnancy (Fig. 1B), but maternal BMI determined in the first trimester was similar in both pregnancies Fig. 1C. We further observed a strong trend towards less obstetric complications in second pregnancy (Fig. 1D). Estimated fetal weight assessed via fetal ultrasound in second and third trimester indicated an improved intra-uterine growth in second pregnancies, but levels of significance were not reached (Fig. 1E). At birth, we observed a 2.5% increased birth weight in a second pregnancy (Fig. 1 F and G) while gestational length was similar (Fig. 1H). Additionally, second pregnancy neonates exhibit a higher Apgar score at 5 minutes suggesting an improved physical and adaptational health, albeit level of significance was not reached (Fig. 1I). PBMCs obtained from 15 women from each trimester revealed frequencies of CD4 Treg cells as well as their CD73 expression to be similar throughout first and second pregnancy (Fig. 1 J and K). Taken together these clinical data support the observation of less complicated subsequent human pregnancies (19). However, we did not observe an changes in systemic CD4⁺ Treg cell frequencies of human PBMCs. Hence, to suitably study the impact and functional relevance of immune memory, we chose various mouse models to investigate the prevailing local immune environment.

Fig. 1. Less obstetric complications and an improved fetal growth in subsequent human pregnancies. (A-I) Maternal and fetal data from pregnant women in their first and second pregnancy (A), respectively, enrolled in the prospective birth cohort study PRINCE (n=78) conducted at the University Medical Center Hamburg-Eppendorf including birth weight (B) maternal age, (C) maternal BMI, (D) pregnancy complication, (E) estimated fetal weight

assessed via prenatal ultrasound, (F) birth weight, (G) birth weight stratified by neonatal sex, (H) gestation length and (I) APGAR score at 5 minutes. (J, K) Flow cytometry analysis of PBMCs obtained from 15 pregnant women of each trimester in their 1st and 2nd pregnancy, including frequency of CD4⁺ Treg cells (J) along with their CD73 expression (K).

Reviewer's comment #1: The interaction between the immune system and the endocrine system is known to contribute to creating a favourable environment for the fetus. Since several hormones associated with pregnancy, such as hCG and steroid hormones, (as estrogens, FSH and LH), participate in the maintenance of immune homeostasis, the authors should evaluate hormonal asset in pregnant mice with higher and lower number of CD4⁺ mTreg cells.

Authors' response: We thank the reviewer for raising this point. We agree that hormones play a fundamental role in the establishment of fetal tolerance. We did measure progesterone and corticosterone in mice as well as progesterone and oestrogen in human. However, we did not detect any significant changes in hormonal response between first and second pregnancy, albeit corticosterone was always slightly higher in 2nd pregnancy compared to 1st pregnancy, but did not reach levels of significance. In our analysis of human samples, we were even very surprised, that progesterone not only remained unchanged, but was even lower in 2nd pregnancy compared to 1st pregnancy during their second trimester, respectively. Also, oestrogen did not show any changes in the maternal serum. Therefore, we have concluded the beneficial effects observed during subsequent pregnancy are a rather cell-mediated mechanism that happens hormone-independent.

We have amended the result section as follows:

“Consequently, the beneficial effects observed during subsequent pregnancy are based on more cell-mediated mechanisms that happen hormone-independent.”

Reviewer's comment #2: It would be useful for the authors to investigate which molecular targets/signatures possess these CD4⁺ mTreg cells, whether they have a specific gene signature that identify a specific molecular asset for pregnancy-related mTreg.

Authors' response: We thank the reviewer for this valid point. We did try to do that by our experiments shown in Fig. 7, deliberately avoiding the use of scRNA-seq and instead analyse right on protein level using the new method of Infinity Flow – a method combining 250 of overlapping flow cytometry panels and using machine learning to enable the simultaneous analysis of these surface-expressed proteins across individual cells.

We started to validate the markers identified in this set of experiments in additional in-vivo experiments as shown below, but since the validation of the various markers will require an extensive evaluation, individually and in combination, it would exceed the scope of the present manuscript. We have discussed the identified candidates and their potential in line 434-473.

Reviewer's comment #3: To complete the analysis of the CD4⁺ mTreg compartment, the authors could assess the status of the periphery (spleens) to observe whether there is an increase in CD4⁺ mTreg cells in subsequent pregnancies. This would also allow them to assess immune cell levels at time zero, before any pregnancy, and subsequently.

Authors' response: We did additional analyses to assess CD4⁺ Treg cells in the peripheral compartment such as spleen and PBMCs in mice and in human. However, we could not observe any differences in these organs. We have included additional data in Figure 1 and S1 to demonstrate the necessity to focus on the local feto-maternal interphase to suitably study the impact and functional relevance of immune memory as shown above. We also provided data on CD4⁺ Tregs before and after pregnancy in Figure 2.

Reviewer's comment #4: The authors conducted a comprehensive analysis of the frequencies of CD4⁺ mTreg cells during the postpartum period and the second pregnancy in mice. However, to gain a deeper understanding of the role of these cells, it would be beneficial for the authors to perform a series of functional experiments to assess whether the increase in frequencies observed during the second and third pregnancies is associated with enhanced functionality of these cells. It would be useful for the authors to purify Treg cells and evaluate their suppressive capacity *in vitro*, using CD4⁺ cells from the same mouse from which the Treg cells were isolated and CD4⁺ cells from a non-pregnant mouse, in order to more accurately assess their functionality. Similarly, the authors could strengthen their findings of an increased level of regulatory T cells following a second and third pregnancy by analyzing several markers related to the suppressive function of these cells and assessing whether these Treg cells are stable over time.

Authors' response: We thank the reviewer for this suggestion. Since pregnancy and the processes of maternal immune adaptation are very complex including soluble factors, e.g. hormones, we are cautious about the informative value of *in-vitro* assays. That is the main reason, we did a couple of functional experiments *in-vivo*, including the depletion and supplementation of Tregs as well as to challenge the "immunological system" with prenatal stress. Our results clearly show, that CD4⁺ Treg cells exhibit an enhanced functionality in subsequent pregnancy as demonstrated by enhanced IL-10 in early pregnancy and improved

pregnancy outcome, e.g. reduced abortion rates and absence of intra-uterine growth retardation after prenatal stress.

Reviewer's comment #5: In order to gain a deeper insight into the inflammatory processes that are known to occur during pregnancy and that could potentially contribute to miscarriage, the authors should perform measurement of circulating cytokine levels to correlate with increased frequencies of Treg cells that they observed (IL10, TGFb, etc).

Authors' response: We thank the reviewer for this consideration. Detection of circulating cytokine levels has often been reported as challenging due to the rapid degradation. Therefore, we decided to use Fig/Tiger mice for our experiment to assess IL-10 in combination with the CD4 Tregs ex vivo. These mice were beneficial due to the two reporters for FoxP3 and IL-10 which allowed the immediate and simultaneous detection of FoxP3 (red fluorescent) and IL-10 (green fluorescent) by flow cytometry without antibody labelling or other experimental procedures like permeabilizing the cell membrane, which could influence the results.

Reviewer's comment #6: To gain a deeper insight into the percentage of mTreg cells illustrated in Figure 1 G and K, the authors may wish to consider selecting a more representative flow cytometry plot and, most importantly, presenting the graphs as violin plots, with the individual percentage values obtained from the different mice.

Authors' response: We thank the reviewer for this suggestion. We have revised the figure accordingly.

Reviewer's comment #7: In Figure 2 I and K, it would be useful to include the values corresponding to gd 3.5, if they are too low, the authors might consider splitting the y-axis in this case.

Authors' response: We thank the reviewer for this suggestion. However, we refrained from include a DC panel in our experiments on gestation day 3.5 due to the consistency of the data

generated using tissue taken on gestation day 7.5 and 15.5. Therefore, we refrain from repeating this experiment on gestation day 3.5 in the context of the revision in order to align to the 3R Principle and limit the number of research animals to an absolute minimum. However, in order to add additional information on DC phenotype and function, we amended the presented data on IL-10 (Fig. 2I) and CD80/86 expression (Fig. 2K) with the time point before the respective pregnancy (virgin vs interpregnancy interval), marked in the figure as pre(-pregnancy).

Reviewer’s comment #8: In Figure 2E, the authors show the percentage of CD4⁺ FoxP3⁺ cells (Treg cells). The violin plot shows very low values, the highest value is about 1%. To be consistent and maintain the same style as the other figures, it would be useful to show a cytofluorimetry plot representing these percentages.

Authors’ response: We thank the reviewer for this suggestion. The issue here lies in the respective parent populations used. While the frequency of regulatory Treg in the violin plots is shown as percentages in all CD45 leukocytes, the respective dot plots have CD4 as parent population. Hence, the numbers written in the dot plots are higher than in the violin plots. In contrast, the percentage of IL-10 is given within CD4⁺ Treg cells in the violin plots, consequently, the numbers written in the dot plots are lower than in the violin plots. Since we did not want to overwhelm the reader with too many dot plots, this was used as a compromise to show both, CD4 Treg cells and IL-10, in one plot. However, we can understand the confusion and therefore, we have provided additional dot plots in Fig. S1 (C and D) and amended the figure legend as follows:

“Pseudocolor plots and respective numbers presented correspond to CD4 as parent population. Additional plots are shown in Fig. S1 D and E”

Reviewer #3:

General comment from this reviewer: In this study, the authors investigated memory T cell function in pregnancy. Using a high-throughput single-cell quantification method, authors found candidate markers such as CXCR4 and CD274 for detecting CD4⁺ mTreg cells. It is reported that findings may contribute to the improved understanding of pregnancy-induced immune memory and foster the identification of immune targets aiming to reduce the risk for immune-mediated pregnancy complications.

The study is well-designed to explain the fundamental questions for the mTreg cells in pregnancy using a murine model. This work is a significant addition to the field of reproductive immunology and obstetrics and gynaecology. The work supports the conclusion and claims. The methodology is sound, and there is a clear explanation for the limitations of the study. The methodology is well written for the work to be reproduced.

Authors' response: We thank the reviewer for the favourable assessment of our manuscript.

Reviewer's comment to:Figure 1:

Reviewer's comment #1: The increased mTregs are limited to the draining LN but not in the uterus: The frequencies of CD44⁺⁺ /CD4 Treg remained unaffected in the uterus with the increasing number of pregnancies. Please explain this finding further in the discussion.

Authors' response: We thank the reviewer for raising this important point. Memory T cells are preferentially located in secondary lymphoid organs, mainly in lymph nodes¹². On the one hand, this may explain the lack of an increase in mTregs in the uterus, on the other hand, of course, we know that there are tissue-resident memory T cells in the uterus e.g. after infection and presumably also after pregnancy. We attempted to compare uterine Treg cells from virgin and mTregs from parous as done with the lymph node (Fig. 7). However, the low number of T cells in a virgin uterus and even lower number of CD4 Treg cells prevented a reliable analysis.

Reviewer's comment #2: The title of Figure 1 can be "Improved neonatal outcome and mTreg and Treg cells in subsequent pregnancies."

Authors' response: We thank the reviewer for this suggestion. Due to the change done to Figure 1, we have revised the title of Figure 1 to "Fig. 1. Higher frequency of CD4⁺ regulatory T (Treg) cells post-partum with increasing number of pregnancies."

Reviewer's comment #3: Are there any age-related changes in the Treg cell population since the controls were virgin mice since the 2nd and 3rd pregnancies were at an advanced age compared to virgin mice?

Authors' response: We thank the reviewer for this question. We would like to clarify the experimental setup. All 20 mice have been age-matched to exclude age as a confounding factor. That means, we started to mate 5 female mice with 5 male mice, respectively, at the age of 8 weeks (mating 1:1), and kept together during pregnancy. After giving birth, we separate the pups from their parents and documented litter size and neonatal weight of the offspring. Subsequently, we added another female to each cage (mating 1:2). After the female mice gave birth again, we added a third female to the cage (mating 1:3) and waited till all dams have given birth. The analysis was performed 4 weeks after the last delivery. Hence, all mice have been non-pregnant at the time of analysis and the three groups were compared to virgin mice which have been left undisturbed during the entire time period.

In order to clarify that, we have amended the text as follows:

"Age-matched female mice were allogeneically mated once, twice or three times, respectively. Four weeks after the last delivery, parous mice were compared to virgin mice, that have been

left undisturbed during the entire time period (Fig. 1A). Equally to our observation in human PBMCs, we detected similar CD4⁺ Treg cell frequencies in murine PBMCs four weeks postpartum, independent on the number of pregnancies. (Fig. 1, B and C, Fig. S1, A and B).”

Reviewer’s comment #4: The litter size and neonatal weight were increased with the increasing orders of pregnancies. However, in LN, there were no differences regarding Treg cells between 2nd and 3rd pregnancies, although there seems to be a stepwise increase in mTreg cells. What is a possible explanation for this finding? If the litter size, neonatal weight, and Treg cells keep increasing with the advanced number of pregnancies, it can be related to other concerns. Please discuss.

Authors’ response: We thank the reviewer for this question. The absent difference regarding Treg cells between 2nd and 3rd pregnancies is a matter of speculation. It could be that a sufficient amount of mTregs are present in the lymph node and a further increase would not be beneficial, since it is always the same fetal (=paternal) antigen. Also, regarding neonatal outcome, we see, that there is no further increase in litter size, probably to avoid exceeding the capacity limit. A change in paternity for a third pregnancy could shed light to that question, in order to see if there is an increase in Tregs, because “two sets of mTregs” must be maintained.

Reviewer’s comment to Figure 2:

Reviewer’s comment #1: Please mark Figure 2, D as LN and Figure 2, G as uterus.

Authors’ response: We thank the reviewer for this comment. We have revised the figure accordingly.

Reviewer’s comment #2: In the Figure 2 legend, please delete the space in H. I,J. Two-way-Anova should be Two-way ANOVA.

Authors’ response: We have revised the figure legend accordingly.

Reviewer’s comment #3: No DC in 3.5 weeks of gestation during 1 and 2nd pregnancy? Please consistently number the figures: Authors describe “Fig 2B-D” or “Fig 3, H-J.”

Authors’ response: We thank the reviewer for this advice. We have checked the references to the figures throughout the manuscript. Unfortunately, we did not include a DC panel in our experiments performed on gestation day 3.5. Since the results determined on gestation day 7.5 and 15.5 were consistent and conclusive, we refrain from a repeat of gestation day 3.5 due to the 3R Principle to limit the number of animals (Reduction) and their suffering (Refinement) in tests to an absolute minimum. As already stated above, in order to add additional information, we provided data on IL-10 (Fig. 2I) and CD80/86 expression (Fig. 2K) before the respective pregnancy (virgin vs interpregnancy interval), marked in the figure as pre(-pregnancy).

Reviewer's comment #4: In Figure 2B, C, E, F, H, I and K, please add a legend to explain yellow and green violin plots

Authors' response: The colors used for the different experimental groups are always explained by the experimental setup (Fig. 2 A)

Reviewer's comment #5: Serum progesterone level goes up with the advance of pregnancies. In this study, serum progesterone level was decreased at 7.5 compared to 3.5 gd.

Authors' response: The decrease in serum progesterone level at gd 7.5 compared to gd 3.5 is not significant. Further, we must be aware, that these are different mice analysed at different time points and not a longitudinal assessment in the same mice. Hence, a direct comparison between gestational days must be interpreted with caution. The primary comparison should be between 1st and 2nd pregnancy.

Reviewer's comment to Figure 3:

Reviewer's comment #1: Figure 3. Please specify that Sox2 was significantly increased but not for Nanog (line 144). Again, labyrinth changes are not statistically significant.

Authors' response: We thank the reviewer for this comment. We have revised the text accordingly:

"We also assessed key parameter of fetal outcome in our mouse models, e.g., the expression of the transcription factors Sox2 and Nanog, which are indicative for timely fetal development early during pregnancy, on gd 7.5 by confocal microscopy (39, 40). Here, we observed higher expression levels of Sox2 and Nanog in fetuses in second, compared to first pregnancies, albeit only significant for SOX-2 (Fig. 3, B-D) which may indicate an advanced early fetal development."

Reviewer's comment #2: Is Figure 3D a staining of the whole fetus? Please explain the staining outcomes in detail.

Authors' response: We thank the reviewer for this comment. Yes, to investigate early fetal parameter, the whole embryos were carefully removed from the uterus and subsequently stained with NANOG and SOX-2 antibodies and analyzed using confocal microscopy as described in detail in the M&M section. NANOG and SOX-2 are both transcription factors for the self-renewal of undifferentiated embryonic stem cells, expressed by embryonic epiblast and stem cells, and allow the quantification of cells in confocal microscopy. The observed

increase of both markers, albeit only significant for SOX-2, indicate an early advanced fetal development.

Hence, we have amended the results sections as follows:

“We also assessed key parameter of fetal outcome in our mouse models (Fig. 3A), e.g., the expression of the transcription factors Sox2 and Nanog, which are indicative for timely fetal development early during pregnancy, on gd 7.5 by confocal microscopy^{3,5}. Here, we observed higher expression levels of Sox2 and Nanog in fetuses in second, compared to first pregnancies, albeit only significant for SOX-2 (Fig. 3, B-D) which may indicate an advanced early fetal development.”

Reviewer’s comment #3: In Figure 3 J, green and blue lines are hard to see. It is recommended to separate the tracing of these lines without background histopathologies. The tracing can be added to each placental cut section.

Authors’ response: We retraced the green and blue lines with a larger line thickness and we hope that the two placental areas are now clearly recognisable.

Reviewer’s comments to Figure 4:

Reviewer’s comment #1:*In Figure 4, please add the explanation for AT. In the legend, 150.000 CD4+ Treg should be 150,000 CD4+ Treg. 4J and L. What does P stand for in J, L, N, and P?

*“student t-test” should be Student t-test.

Authors’ response: We thank the reviewer for these points. We have revised the figure (P = Pregnancy) and the figure legend accordingly.

Fig. 4. Modulation of CD4⁺ regulatory T (Treg) cell number and antigen-specificity during pregnancies.

(A) Experimental setup: age-matched DEREK mice were allogeneically mated to Balb/c males once or twice, respectively. Some second pregnancy mice were injected with diphtheria toxin (DT) after the first delivery to deplete CD4⁺ Treg cells and subjected to a recovery phase of 14 days before re-mating. On gestational day (gd) 15.5, frequencies of CD4⁺ Treg cells in (B and C) lymph node and (D) uterus were assessed by flow cytometry. Data for gd 7.5 are shown in fig. S1. (E) Experimental setup: age-matched Fir/Tiger mice were allogeneically mated to OVA-Balb/c males. Within 7 days postpartum, CD4⁺ Treg cells from lymph node and uterus were harvested from parous mice (termed memory (m)Tregs). Simultaneously, CD4⁺ Treg cells harvested from virgin mice served as controls (termed naïve (n)Tregs). Subsequently, 150,000 CD4⁺ Treg cells were adoptively transferred (AT) into first pregnancy mice mated to OVA-Balb/c males on the day of plug. (F) Histograms of fetal antigen (OVA)-specific CD4⁺ Treg cells in first (black line) and second (blue line) in lymph node (LN, top) and uterus (bottom). Numbers represent the frequency of OVA⁺ cells in the CD4⁺ Treg cell compartment. Flow cytometry analysis of first pregnancy CD4⁺ Treg frequencies was performed in (G) lymph node and (H) uterus samples. (I) Experimental setup: age-matched Fir/Tiger mice were allogeneically mated to Balb/c males once or twice, respectively. Additionally, some mice were mated to DBA mice for their second pregnancy (DF = different father). On gd 15.5, frequencies of CD4⁺ Treg cells in (J and K) lymph node and (L) uterus were assessed by flow cytometry. Data regarding CD44^{high} and IL-10 expression are shown in fig. S1. (M) Experimental setup: age-matched Fir/Tiger mice were allogeneically mated to Balb/c males once or twice, respectively. Additionally, first and second pregnancy mice were prenatally sound-stressed mid-gestationally on gd 10.5, 12.5 and 14.5. On gd 15.5, flow cytometry analysis assessed CD4⁺ Treg cell frequencies in (N and O) lymph node and (P) uterus. Maternal serum progesterone levels are shown in fig. S1) Data are presented as violin plots with individual point, median and quartiles, and the statistical significance between groups was calculated using Student’s t-test when comparing two groups or One-way ANOVA when comparing three or more groups (* p < 0.05, ** p < 0.01).

Reviewer’s comment #2: nTreg was adoptively transferred, uterine or LN Tregs did not increase. Is there any possibility that allo rejection may play a role in this finding?

Authors’ response: Although we cannot totally exclude allo rejection, we are confident that this is not the case here. The cells that were adoptively transferred were of the same

background (C57Bl/6J) as the female mice. Further, allo-rejection should have happened also with the mTregs that were transferred.

Reviewer's comments to Figure 5:

Reviewer's comment #1: Figure 5 B. Was the abortion rate 0% for not depleted animals?

Authors' response: Yes, indeed. Please see also Figure 3 F. We rarely observed any abortion in 2nd pregnancy, while a 1st pregnancies showed an average abortion rate of 10% in our animal facility. Due to the depletion of the CD4⁺ (memory) Tregs during the interpregnancy interval, the abortion rate in 2nd pregnancy returned to 1st pregnancy-level.

Reviewer's comment #2: Line 239. Prenatal stress was given on gd 10.5, 12.5 and 14.5. In the graph, it is listed as gd 15.5. Line 225. The authors stated that the abortion rate was not changed as expected due to the late stress exposure during gestation. What is the abortion rate when the stress was given on gd 10.5 or 12.5? In the figure legend, it seems that experiments were also performed on those days.

Authors' response: We are sorry for the confusion. Our prenatal sound stress model includes the exposure to sound stress three times every other day starting on gestation day 10.5 – always for 24 h with subsequently 24 break – using a rodent repellent device (Conrad Electronics) emitting sound signals at a frequency of 460 Hz and with an intensity of 88 dB lasting 1 second at randomly spaced intervals which occur four times per minute which has been placed into the cage ⁷. Gestational day 15.5 – after the last stress exposure – the mice have been sacrificed for analysis.

Since on gestation day 10.5 pregnancy is already firmly established, an increase in abortion rate was not expected but rather an effect on fetal growth.

Reviewer's comments to Figure 6:

Reviewer's comment #1: In 2nd pregnancy, the % ExIL17 and Tr1 cell proportion were significantly increased compared to the first pregnancy. It is postulated that these changes may be induced upon implantation or labor-related inflammation during the first pregnancy. Th17 cells increase during these conditions. Is there any other evidence that increased Th17 cells induce transdifferentiation to Treg cells?

Authors' response: We are not sure, we understand the question correctly. However, the Th17/ Treg balance and their transdifferentiation, especially during pregnancy, remains poorly understood. We know from other setting, that Treg plasticity contributes to resolution of inflammation, e.g. in the gut. Hence, we hypothesized that this plasticity might also have an impact in pregnancy as a tight balance of inflammatory and anti-inflammatory events needs to be controlled in the uterus, due to e.g. periodic remodelling of the uterine tissue or during pregnancy (implantation – immune tolerance – birth). Our data indicate that a transdifferentiation of Th17 cells into Tregs might be an independent pathway to support immune adaptation and fetal outcome in subsequent pregnancies. However, that needs further investigation. We have addressed that aspect in the discussion:

“We here introduce the transdifferentiation of Th17 cells to CD4⁺ Treg cells as a potential independent mechanism to improve maternal immune adaptation and pregnancy outcome in subsequent pregnancies. The uterus is a unique organ that is characterized by profound tissue remodeling, local inflammation due to infection and phases of substantial inflammation during pregnancy (54, 71). Hence, the balance of inflammatory and anti-inflammatory events could be maintained by the transdifferentiation of Th17 cells to CD4⁺ Treg. However, this Th17/ Treg balance, especially during pregnancy, remains poorly understood (72, 73) and further studies are necessary to unveil the underlying mechanisms.”

Reviewer's comment #2: Figure 6B. Please add legends for different color violin plots.

Authors' response: We thank the reviewer for this point, but the color for the violin plots is indicated in the experimental setup displayed in Figure 6A. We amended Figure 6A to clearly allocate the respective groups to the respective colors.

Reviewer's comments to Figure 7:

Reviewer's comment #1: *In Fig 7 G, Please add the MFI legend displayed.

Authors' response: We apologize for that oversight and have added the legend to Figure 7 G.

Reviewer's comments to the Discussion:

Reviewer's comment #1: A lack of TRM cells in the uterus was explained as a lack of proper markers. The difference between the uterus and other organs, such as the skin or small intestine, is menstruating features. While the functional layer of the endometrium is exfoliating, Where is the actual location of TRM cells in the endometrium or decidua spacially?

Authors' response: We thank the reviewer for raising this important point. We have address that in the discussion:

"This includes the actual location of T_{RM} cells in the endometrium or the decidua which could by identified by spatial transcriptomics, mapping the whole transcriptome or targeted gene expression to specific locations in a tissue."

Reviewer's comment #2: Line 353-360: Endometrial TRM cells also express CD49a, CCR5, and PD1. CCR5 expression was related to IL17 production. Recently, tissue-resident memory $\alpha\alpha\alpha T$ cells have been reported in the murine uterus. This section can be enhanced by updating the reported data regarding TRM cells in the uterus. Lines 421-430: The possible impact of over-expansion of Treg and mTreg cells should be considered regarding the advanced number of pregnancies.

Authors' response: We were not able to find the mentioned publication. However, we have advanced the discussion as follows:

"Further, the ability of CD4⁺ Treg cells to remain tissue resident long-term is still rather controversial. So called "tissue Tregs" are predominantly investigated in visceral adipose tissue where they control local inflammation (55). It was shown that inflammation-experienced "memory" Tregs preferentially localized in non-lymphoid tissue, partially caused by the expression of CXCR3 (56). Further, CD4⁺ Treg cells in the female reproductive tract were found to be more activated compared to circulating CD4⁺ Treg cells, indicated by higher expression of ICOS, TIGIT, CD39, CTLA-4, and GITR, suggesting also higher suppressive capacity (57). Hence, the interplay of uterine T_{RM} cells and memory CD4⁺ Treg cells might be crucial to balance immune surveillance and foetal tolerance and the functional role of tissue-resident CD4⁺ mTreg cells in ameliorating the outcome of second pregnancy needs further clarification."

References

1. Clark, D.A. (2016). The importance of being a regulatory T cell in pregnancy. *Journal of Reproductive Immunology* 116, 60–69. <https://doi.org/10.1016/j.jri.2016.04.288>.
2. Rowe, J.H., Ertelt, J.M., Xin, L., and Way, S.S. (2012). Pregnancy imprints regulatory memory that sustains anergy to fetal antigen. *Nature* 490, 102–106.
3. Zhang, S., and Cui, W. (2014). Sox2, a key factor in the regulation of pluripotency and neural differentiation. *World J Stem Cells* 6, 305–311. <https://doi.org/10.4252/wjsc.v6.i3.305>.
4. Shao, T.-Y., Kinder, J.M., Harper, G., Pham, G., Peng, Y., Liu, J., Gregory, E.J., Sherman, B.E., Wu, Y., Iten, A.E., et al. (2023). Reproductive outcomes after pregnancy-induced displacement of preexisting microchimeric cells. *Science* 381, 1324–1330. <https://doi.org/10.1126/science.adf9325>.
5. Carlin, R., Davis, D., Weiss, M., Schultz, B., and Troyer, D. (2006). Expression of early transcription factors Oct-4, Sox-2 and Nanog by porcine umbilical cord (PUC) matrix cells. *Reproductive Biology and Endocrinology* 4, 8. <https://doi.org/10.1186/1477-7827-4-8>.
6. La Rocca, C., Carbone, F., Longobardi, S., and Matarese, G. (2014). The immunology of pregnancy: Regulatory T cells control maternal immune tolerance toward the fetus. *Immunology Letters* 162, 41–48. <https://doi.org/10.1016/j.imlet.2014.06.013>.
7. Wiczorek, A., Perani, C.V., Nixon, M., Constanica, M., Sandovici, I., Zazara, D.E., Leone, G., Zhang, M.-Z., Arck, P.C., and Solano, M.E. (2019). Sex-specific regulation of stress-induced fetal glucocorticoid surge by the mouse placenta. *American Journal of Physiology-Endocrinology and Metabolism* 317, E109–E120. <https://doi.org/10.1152/ajpendo.00551.2018>.
8. DeJong, C.S., Maurice, N.J., McCartney, S.A., and Prlic, M. (2020). Human Tissue-Resident Memory T Cells in the Maternal-Fetal Interface. Lost Soldiers or Special Forces? *Cells* 9, 2699. <https://doi.org/10.3390/cells9122699>.
9. Saito, S. (2022). Reconsideration of the Role of Regulatory T Cells during Pregnancy: Differential Characteristics of Regulatory T Cells between the Maternal-Fetal Interface and Peripheral Sites and between Early and Late Pregnancy. *Medical Principles and Practice* 31, 403–414. <https://doi.org/10.1159/000527336>.
10. Khantakova, J.N., Bulygin, A.S., and Sennikov, S.V. (2022). The Regulatory-T-Cell Memory Phenotype: What We Know. *Cells* 11. <https://doi.org/10.3390/cells11101687>.
11. Thiele, K., Ahrendt, L.S., Hecher, K., and Arck, P.C. (2019). The mnemonic code of pregnancy: comparative analyses of pregnancy success and complication risk in primary and secondary human pregnancies. *Journal of Reproductive Immunology*.
12. Hamon, M.A., and Quintin, J. (2016). Innate immune memory in mammals. *Semin Immunol* 28, 351–358. <https://doi.org/10.1016/j.smim.2016.05.003>.

Point-by-point reply to reviewers' comments on manuscript entitled 'Pregnancy-acquired memory CD4⁺ regulatory T cells improve pregnancy outcome in mice' by K. Thiele *et al.* (NCOMMS-24-48524A)

Reviewer comments:

Reviewer #1 and #2:

General comment from the reviewers: The paper has overall been improved and will be a nice contribution to the field.

Authors' response: We thank the reviewers for this final evaluation to our revised manuscript.

Reviewer #3:

General comment from this reviewer: The authors have provided a thorough and well-structured response to the reviewers' comments and most concerns and suggestions have been adequately addressed with appropriate revisions and additional experiments.

Authors' response: We thank the reviewers for this final evaluation to our revised manuscript. Please find below our response regarding the one last concern raised by the reviewer.

Reviewer's comment #1: The response regarding Figure 1E is noted. While the authors reported improved intrauterine growth in the second and third trimesters of subsequent pregnancies, data from the first pregnancy are missing from the plot. It would be helpful for the authors to clarify whether these data are available and, if not, provide further justification.

Authors' response: I am actually not sure, what the reviewer is referring to. Figure 1E shows the trajectory of estimated fetal weight assessed via prenatal ultrasound in all three trimesters. There are not data missing, the graph is showing the curves for 1st pregnancy (beige) and 2nd pregnancy (turquoise). I can see that the graphs are quite overlapping, but there are both presented for all three trimesters. In order to improve clarity, we have amended the result section with the precise values:

"Estimated fetal weight assessed via fetal ultrasound in second and third trimester indicated an improved intra-uterine growth in second pregnancies, but levels of significance were not reached (Fig. 1e, mean±SD of 1st versus (vs) 2nd pregnancy: 1st trimester: 103.0g ± 21.8g vs 98.80g ± 13.43g, 2nd trimester: 647.5g ± 77.9g vs 660.8g ± 77.1g, 3rd trimester: 2625g ± 322.4g vs 2706g ± 356.1g)."